# Novel QTLs for salinity tolerance revealed by genome-wide association studies of biomass, chlorophyll and tissue ion content in 176 rice landraces from Bangladesh

**Md Nafis Ul Alam**[1], **G. M. Nurnabi Azad Jewel**[1,2]**, Tomalika Azim**[1]**, Zeba I. Seraj**[1]*

**1** Department of Biochemistry and Molecular Biology, University of Dhaka, Dhaka, Bangladesh,
**2** Department of Genetic Engineering and Biotechnology, Shahjalal University of Science and Technology, Sylhet, Bangladesh

* zebai@du.ac.bd

**Data Availability Statement:** All data and source code can be found at https://github.com/DeadlineWasYesterday/Cat-Does-Plant/.

## Abstract

Farmland is on the decline and worldwide food security is at risk. Rice is the staple of choice for over half the Earth's people. To sustain current demands and ascertain a food secure future, substandard farmland affected by abiotic stresses must be utilized. For rapid crop improvement, a broader understanding of polygenic traits like stress tolerance and crop yield is indispensable. To this end, the hidden diversity of resilient and neglected wild varieties must be traced back to their genetic roots. In this study, we separately assayed 11 phenotypes in a panel of 176 diverse accessions predominantly comprised of local landraces from Bangladesh. We compiled high resolution sequence data for these accessions. We collectively studied the ties between the observed phenotypic differences and the examined additive genetic effects underlying these variations. We applied a fixed effect model to associate phenotypes with genotypes on a genomic scale. Discovered QTLs were mapped to known genes. Our explorations yielded 13 QTLs related to various traits in multiple trait classes. 10 identified QTLs were equivalent to findings from previous studies. Integrative analysis assumes potential novel functionality for a number of candidate genes. These findings will usher novel avenues for the bioengineering of high yielding crops of the future fortified with genetic defenses against abiotic stressors.

## 1. Introduction

10 million hectares of arable land is lost to urbanization every year [1]. An accelerated wave of climate change damages the quality of the remaining landmass [2]. About half of the world's currently estimated farmland is affected by abiotic stresses, most notably, salinity [3, 4]. Among cereal crops, rice is the most sensitive to abiotic factors and yield losses [5] but has the largest contribution to global food production. Rice feeds more people across the world than any other crop [6], accounting for up to 80% of daily calorie intakes of half the world's population [7]. It is surmised that global rice production has to double by the year 2050 to combat

**Funding:** The author(s) received no specific funding for this work.

**Competing interests:** The authors have declared that no competing interests exist.

these burdens imposed on food security [8, 9]. The insufficiency in finite arable land has to be overcome by increases in crop productivity. This must involve the introgression and propagation of stress tolerance traits to enable farming in soil less ideal for present day elite varieties.

With the help of high-resolution genotyping technologies, it has become possible to elucidate the molecular signature of complex polygenic traits. Multitudes of small genetic effects and their interactions with the surroundings underlie the quantitative assertion of these phenotypes. This limits our ability to refine them through direct breeding or molecular approaches. Many statistical and genome-wide association based studies (GWAS) have been carried out for salinity tolerance [10, 11], chlorophyll [12] and yield [13] related traits in rice independently. Researchers often point out that complex traits are interlinked through their evolutionary origins and must be studied in conjunction to reach conclusive outcomes [14]. Few recent GWAS studies explore abiotic stress tolerance in relation to developmental and agronomic phenotypes in rice. Identification of major-effect QTLs for yield, tolerance and developmental attributes will continue to add layers of sophistication to the current understanding of plant biology. Characterization of causal polymorphisms will facilitate the introgression of salient features into elite varieties with the help of sophisticated molecular technologies such as CRISPR-Cas9 genome editing.

Bangladesh is the largest delta on the planet, endowed with fertile land, meandering rivers and a pronounced history of growing rice. To accelerate the improvements in cereal crop development, we phenotyped 176 local rice varieties from Bangladesh for 11 quantitative traits relating to plant biomass, chlorophyll content, tissue ion content and visual salt damage (SES). We incorporated next-generation sequencing data with high genomic coverage to estimate additive genetic effects and correlate the observed phenotypic values with genomic predictions. We tested over 4 million high-quality SNP markers for association signals separately for each trait. We cross-referenced our identified QTLs with functional annotations, publicly available gene expression profiles and allelic substitution effects. The compiled assessment of multiple traits belonging to several trait classes accentuated the genetic landscape influencing their effects. For identified QTLs, integrative analysis of auxiliary data enabled us to unravel potential novel gene roles.

## 2. Methods

### 2.1 Data and source code

All data, source code and shell commands are documented on GitHub at https://www.github.com/DeadlineWasYesterday/Cat-does-plant. The complete computational pipeline was executed in an HP Z840 workstation running on a 16-core Intel Xeon processor with 256GB of RAM.

### 2.2 Plant material and growth conditions

The 176 rice accessions were received from the IRRI seed bank to ascertain their identity with the 3,000-rice genome project [15]. Seeds were multiplied at BRRI (Bangladesh Rice Research Institute) fields in the months of March and April when the average day/night temperature was 32˚/28˚C with an average humidity of 72%. S1 Table lists the accession codes and metadata with local names and subpopulation information for all 176 varieties. Only 3 varieties belong to the USDA core (IRGC 31727, IRGC 27555 and IRGC 58736) and none are listed in the USDA mini-core collection. 106 accessions are traditional landraces that are cultivated locally in Bangladesh. The remaining 70 accessions are varieties that had been developed by BRRI from local landraces for local agronomy. With the exception of three (IRGC 49375, IRGC 126002 and IRGC 124432), all varieties had been originally collected from Bangladesh

by the International Rice Research Institute. The starter template for the map of Bangladesh was collected from Vecteezy [16] and modified in Adobe Illustrator. Geolocations were labelled using the ggplot2 and ggrepel libraries in R.

The phenotype screening for salinity tolerance at the seedling stage took place in a nethouse enclosure at an average day/night temperature of 31˚C/27˚C and approximately 72% relative humidity. The screening for salinity tolerance at seedling stage was carried out following the methods described by Amin et al. 2012 [17]. Sprouted seeds were sown in netted Styrofoam and floated in 3×3 (3 for control and 3 for stress) replicated PVC trays containing 60L Yoshida solution [18]. The positions of the 176 accession seedlings were randomized in each of the 6 trays. The germinated seeds were allowed to grow for 14 days. Then, NaCl stress was applied gradually starting from 4dS/m up to 12dS/m by 2dS/m increments every 24 hours in all of the experiment trays. The solution level of the trays were maintained with water. Phenotypic measurements were taken after 16 days of stress application, when 90% of the leaves of the sensitive control (IR29) were damaged.

## 2.3 Phenotyping

After about 1 month of germination and 16 days from the first salt stress exposure for the stress condition plants, seedlings were systematically phenotyped. Fresh weight and length measurements were taken for whole roots and shoots. For root and shoot ion content data, plants were externally washed and oven dried at 70˚C. Ground-up dried roots and leaves were treated with 1N HCl for 48 hours before being assayed in a flame photometer (Sherwood model 410, Sherwood, UK). Ion measurements are denoted in millimolar concentration. SES (Standard Evaluation System) scores were given to each plant on the basis of growth stunting, damage to leaves, leaf chlorosis, drying of biomass and plant vigor as described by Gregorio et al. 1997 [19, 20]. Chlorophyll content was measured in fully expanded third leaves. Leaf extracts of 1cm$^2$ size were dissolved in 80% acetone in the absence of light. Absorbance at 645nm and 663nm wavelengths were measured. The formulae for chlorophyll A and B estimation were adapted from Yoshida et al. 1976 [18] as follows:

$$Chlorophyll\ A\ content = (12.7 \times A_{663} - 2.69 \times A_{645}) \times \frac{V}{1000 \times W}$$

$$Chlorophyll\ B\ content = (22.9 \times A_{663} - 4.68 \times A_{645}) \times \frac{V}{1000 \times W}$$

Here $A_{663}$ and $A_{645}$ refer to the absorbance at 663nm and 645nm wavelengths respectively. $V$ is volume in milliliters, $W$ is the fresh weight of the leaf pieces in milligrams and chlorophyll content is expressed as milligrams per gram of fresh leaf tissue.

## 2.4 Phenotype statistics

We have catalogued 10 basic traits that we grossly assign to plant biomass, chlorophyll content and tissue ion content categories. The 11th basic trait is the visual SES score for evaluating salt damage which by definition is only measured in stress condition. Simple mathematical operations on the phenotypes gave rise to 12 additional derived phenotypes. For the 10 traits measured in both conditions, heritability was measured by the formula $h2 = \sigma_G^2 / \left( \sigma_G^2 + \frac{\sigma_e^2}{n} \right)$ where $\sigma_G^2$ is the variance of the genotypes, $\sigma_e^2$ is the residual variance and $n$ is the number of replicates. The variance components were calculated by including the genotype, treatment condition and interaction between genotype and condition as random effects in the *lmer* function from the

lme4 package in R. Marker-based heritability was calculated using the kinship matrix K (details in section 2.7) in the *marker_h2* function from the heritability package in R. Broad-sense heritability was calculated by dividing the variance of the genotype means by the overall phenotypic variance. Correlation and GWAS studies involved phenotype means for three replicates. Linear correlation is expressed by the Pearson's correlation coefficient. Genomic prediction for SNP effects on phenotype was calculated by ridge regression using the *mixed.solve* function from the rrBLUP package [21]. Assessment of data distributions was carried out by the SciPy module in Python. Phenotype transformation for GWAS was done by an ideal genotype derived warping function calculated by WarpedLMM [22].

## 2.5 Preprocessing genotype data

Next-generation sequencing data was compiled from the 3,000-rice genome project [15]. In the original study, libraries were created from young leaves and sequenced in the Illumina HiSeq2000 platform to generate 90bp paired-end reads. Raw sequence reads were aligned to the IRGSP 1.0 reference genome for *Oryza sativa sub. japonica* cultivar Nipponbare. The average sequencing depth was 14× with genome coverage and mapping rates over 90%. Publicly available VCF data from aligned reads were sourced for 176 varieties and compiled using VCFtools in the SAMtools suite. SNPs having phred-scaled quality scores below 30 were flagged as low-quality reads and considered missing data. Beagle 5.1 [23] was used for imputation of unphased genotypes. There were on average 2,930,739 missing genotypes out of 15,241,544 total marker calls per individual with a standard deviation of 492,558. For assessment of imputation accuracy, we prepared a test set having the original proportion of missing data by systematically removing high-quality reads from 1.85 million markers which had no missing reads. Unphased imputation accuracy was measured to be over 99.5% and the report can be found on the git repository. Working files were prepared using in-house python scripts. SNPs that originally had over 20% low quality reads were excluded from all studies. For GWAS, a minor allele frequency filter of 5% was applied to the whole population. Because of the smaller sample size, the minor allele frequency filter for the subpopulations was set at 10%.

## 2.6 Population structure estimation

A combination of methods was employed for deciphering population structure. Principle components were calculated using R and clustered in two and three dimensions respectively using a k-means clustering algorithm. The first three principal components showed 38% explained variation in scree plots. 2D and 3D scatter plots were also plotted in R using the first three principal components and can be seen in the git repository. Bayesian maximum likelihood-based population structure estimation software fastStructure [24] implemented in python was used for assessing admixture between groups. K (number of subpopulations) values from 2 to 15 were tested and the *chooseK.py* function was used to select k = 3. Distances for the neighbor-joining tree were calculated in TASSEL 5 [25] and visualized by the web-based application iTOL (interactive Tree Of Life) [26].

## 2.7 Genome-wide association studies

The primary algorithm for association testing was BLINK [27] in the GAPIT software package [28]. The more popular and statistically robust compressed mixed linear model (CMLM) applied both in GAPIT and TASSEL 5 were also run to validate the findings (data in git repository).

BLINK employs two successive fixed effects models. The first model tests for associations and calculates p values and the second model iteratively improves the first model by exporting markers as covariates using Bayesian Information Criterion (BIC).

The first model is denoted:

$$y_i = S^*_{i1}b_1 + S^*_{i2}b_2 + \ldots + S^*_{ik}b_k + S_{ij}d_j + e_i$$

where $y_i$ is the estimated observation for the $i$-th individual. $S^*_i$ terms represent covariates named pseudo QTNs that start off as an empty set and subsequently are iteratively selected by the second model. $b$ terms are the corresponding effects for the pseudo QTNs. $S_{ij}$ is the genotype for the testing marker $j$. $d_j$ is the effect of the $j$-th marker and $e_i$ is the residual error term having a distribution $e_i \sim N(0, \sigma_e^2)$.

The second model given below is similar but lacks the genotype term:

$$y_i = S^*_{i1}b_1 + S^*_{i2}b_2 + \ldots + S^*_{ik}b_k + e_i$$

Markers are sorted by ascending order of p value and fitted in the second model one by one if considered significant after Bonferroni correction *and* found to not be in linkage disequilibrium ($r^2 > 0.7$) with any marker already included in the equation. Model selection by BIC finds the best fit second model from all combinations and the covariates from that model are exported to the first model.

This complete process is repeated in BLINK until the second model no longer selects new covariates for inclusion in the first model and the first model is established as the final testing model.

To correct for population structure and cryptic relatedness, the kinship matrix K, calculated using GAPIT and the Q matrix having the first three principal components were fit in the first model as covariates. For the compressed mixed linear models fit for validation tests, the optimum compression level was selected and the same covariates were included. Q-Q plots and Manhattan plots were visualized by the qqman package in R [29].

An adjusted Bonferroni correction was applied to set the genome wide significant and suggestive thresholds. The formula used for setting the significance threshold was $-\log(1/M)$ where M is the total number of testing markers. In the whole population, the value was 6.63 for 4,283,120 markers that remained after the MAF filter, which was rounded down to 6.5 for use. In Indica and Aus subpopulations, there were 2,735,794 and 2,562,596 markers respectively and the significant threshold was set at 6.4 accordingly. For suggestive thresholds, effective number of markers were calculated, defined as those being in approximate linkage equilibrium across the genome. A pairwise LD cutoff of $r^2 = 0.2$ with 50kb sliding windows and step size of 50kb in PLINK [30] was used to calculate the effective number of markers. The effective number was 384875, 255487 and 234071 for the whole, Indica and Aus sets respectively. The resulting suggestive threshold values were 5.37, 5.4 and 5.58, from which a common down-rounded value of 5.0 was used.

## 3. Results

### 3.1 Observation of population structure

The panel of 176 accessions predominantly consist of the Indica and Aus subgroups. Fig 1 shows 114 known geolocations from S1 Table where the labelled varieties have been historically cultivated. The appropriate number of subpopulations (K) was estimated to be 3. Three distinct clusters could also be visualized in a scatter plot of the first and second principal components. An admixture plot for 3 subpopulations and the NJ tree for the 176 individuals can

# Local Rice Varieties on an Illustrated Map of Bangladesh

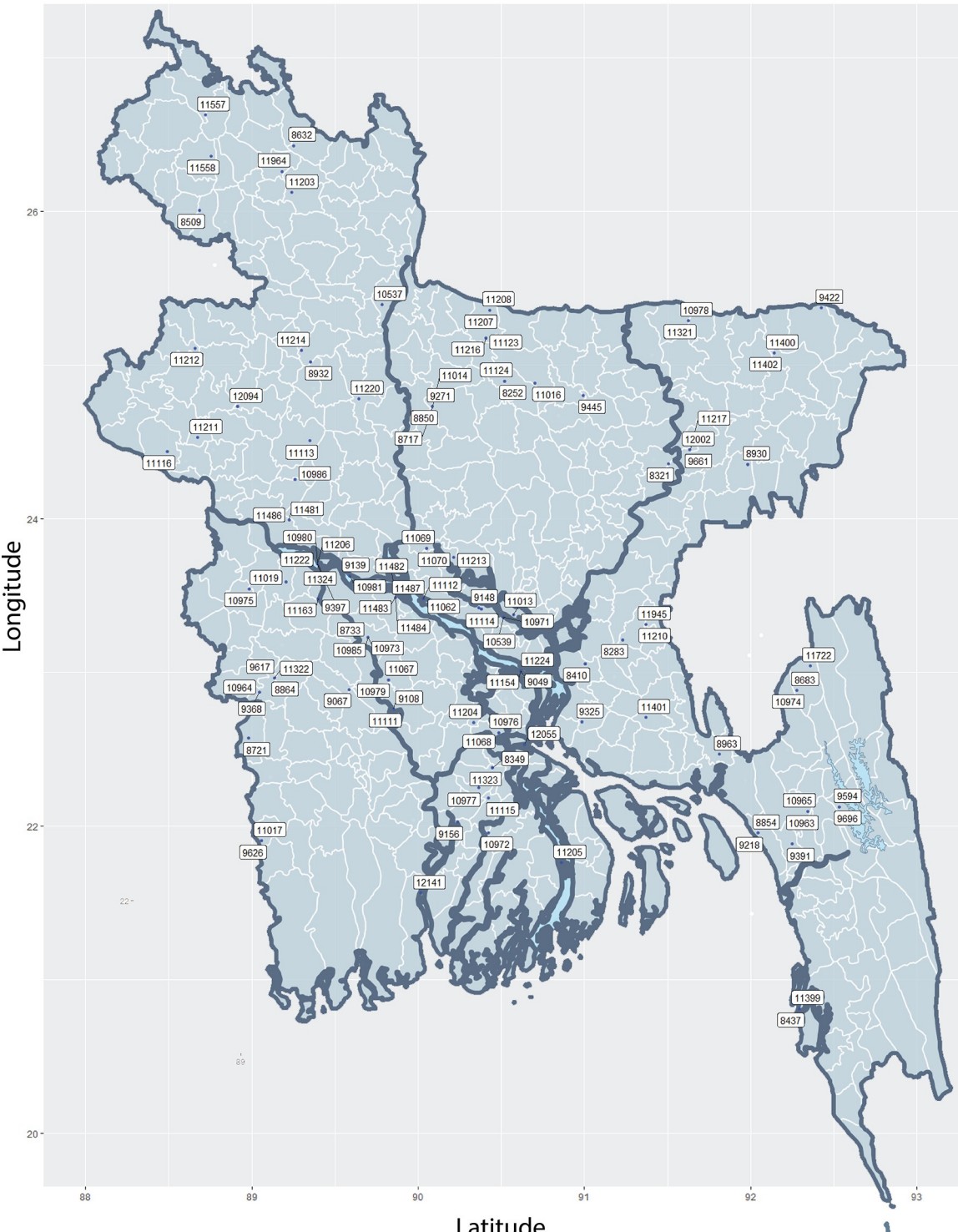

**Fig 1. Map of Bangladesh showing the locations where 114 out of 176 rice varieties from this study have been traditionally propagated and cultivated.** The labels in the image are the IRIS codes from S1 Table lacking the common 'IRIS_313-' prefix.

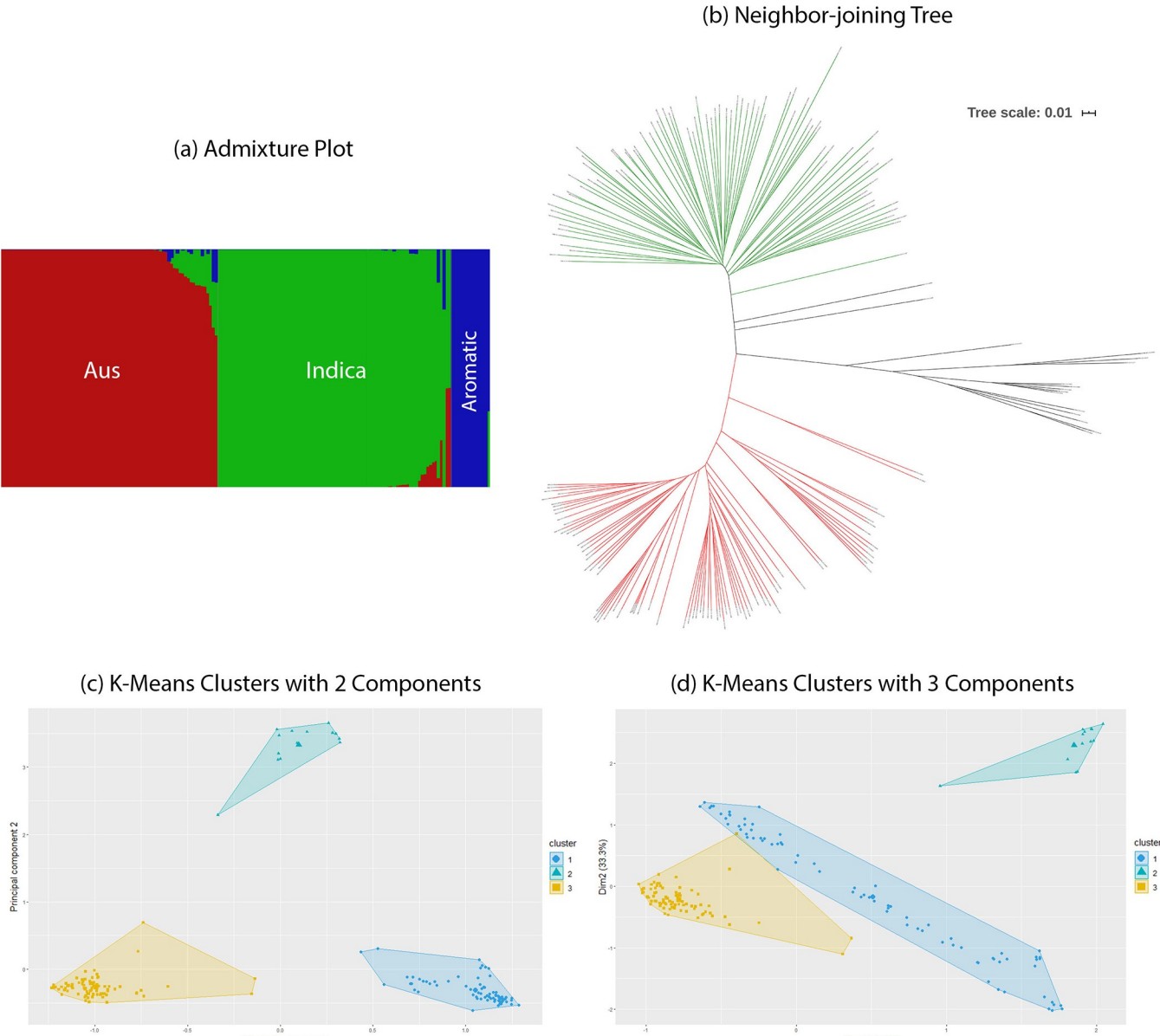

**Fig 2. Population structure estimation results for 176 varieties.** Three distinct subpopulations can be inferred from these diagrams. (a) An admixture plot calculated from fastStructure. (b) Neighbor-joining tree drawn from genotype data. The green individuals belong to the Aus subgroup, red individuals are from the Indica subgroup and black lines denote plants distant from these groups. (c) K-means clustering results of the first two principal components. (d) K-means clustering results of a three-dimensional plot of the first three principal components.

be seen in Fig 2A and 2B respectively. The red individuals belong to the Aus subpopulation and the green shaded individuals are members of the Indica subpopulation. We analyzed 2D and 3D scatter plots for the first three principal components to infer population structure. Bayesian k-means clustering results can be seen in Fig 2C and 2D. The subpopulations assigned to the individuals by fastStructure, the groupings calculated by the k-means algorithm, and that inferred by manual inspection were found to be identical. Population structure inferences from all sources have been tabulated in S1 Table. In the NJ tree and the PCA scatter plots, the Aus subpopulation was found to cluster more tightly than the individuals belonging

to the Indica subgroup. The third subpopulation lodges only a few individuals that are grouped as aromatic rice varieties and are locally known as Basmati or Sadri rice. Because of having a very small number of individuals, the aromatic subpopulation was excluded from subpopulation specific association testing along with three accessions that did not cluster with either subpopulation (S1 Table).

## 3.2 Comprehensive analysis of phenotypes

Eight additional phenotypes could be derived from the biomass traits. Table 1 lists all basic and derived phenotypes with the formulations for each derivation from the basic traits. Biomass traits had lower variance across replicates and were more heritable than ion content traits. Within the biomass category, shoot traits were more heritable than root traits. Marker derived heritability for all phenotypes and broad-sense heritability for applicable phenotypes can be found in S2 Table. The means of phenotype values recorded for every genotype is stored in S3 Table. Narrow-sense heritability could not be calculated from additive marker effects because the grand mean of the phenotypes could explain more variance than the *mixed.solve* prediction model. This could be attributable to the relatively small sample size with diverse and heterozygous wildtype genotypes used for genomic prediction. The linear Pearson's

**Table 1. Basic and derived phenotypes.**

| Group | Basic traits | Heritability of basic traits | Derived traits | Formula for derivation | Heritability of derived traits |
|---|---|---|---|---|---|
| Biomass | Root weight | 0.519935202 | Lost root weight | Root weight in control condition—Root weight in stress condition | 0.020508891 |
| Biomass | Shoot weight | 0.598616239 | Lost shoot weight | Shoot weight in control condition—shoot weight in stress condition | 0.063502564 |
| Biomass | Root length | 0.538476891 | Lost root length | Root length in control condition—Root length in stress condition | 0.186636646 |
| Biomass | Shoot length | 0.680380481 | Lost shoot length | Shoot length in control condition—shoot length in stress condition | 0.00847037 |
| Biomass | | | Root weight per unit length** | Root weight / Root length | 0.441443378 |
| Biomass | | | Shoot weight per unit length** | Shoot weight / Shoot length | 0.234151181 |
| Biomass | | | Lost root weight per unit length | Root weight per unit length in control condition—Root weight per unit length in stress condition | 0.128275455 |
| Biomass | | | Lost shoot weight per unit length | Shoot weight per unit length in control condition—Shoot weight per unit length in stress condition | 0.021422646 |
| Ion content | Root sodium content | 0 | | | |
| Ion content | Root potassium content | 0.015590152 | | | |
| Ion content | Shoot sodium content | 0.029297234 | | | |
| Ion content | Shoot potassium content | 1.05E-09 | | | |
| Chlorophyll content | Chlorophyll A content | 0.299207727 | Lost chlorophyll A content | Chlorophyll A content in control condition—Chlorophyll A content in stress condition | 0.177123042 |
| Chlorophyll content | Chlorophyll B content | 0.273868031 | Lost chlorophyll B content | Chlorophyll B content in control condition—Chlorophyll B content in stress condition | 0.181277873 |
| SES | SES score* | 0.37256389 | | | |

* traits measured in only one condition.

** abbreviated as 'thickness' throughout the manuscript and applicable to both control and stress conditions.

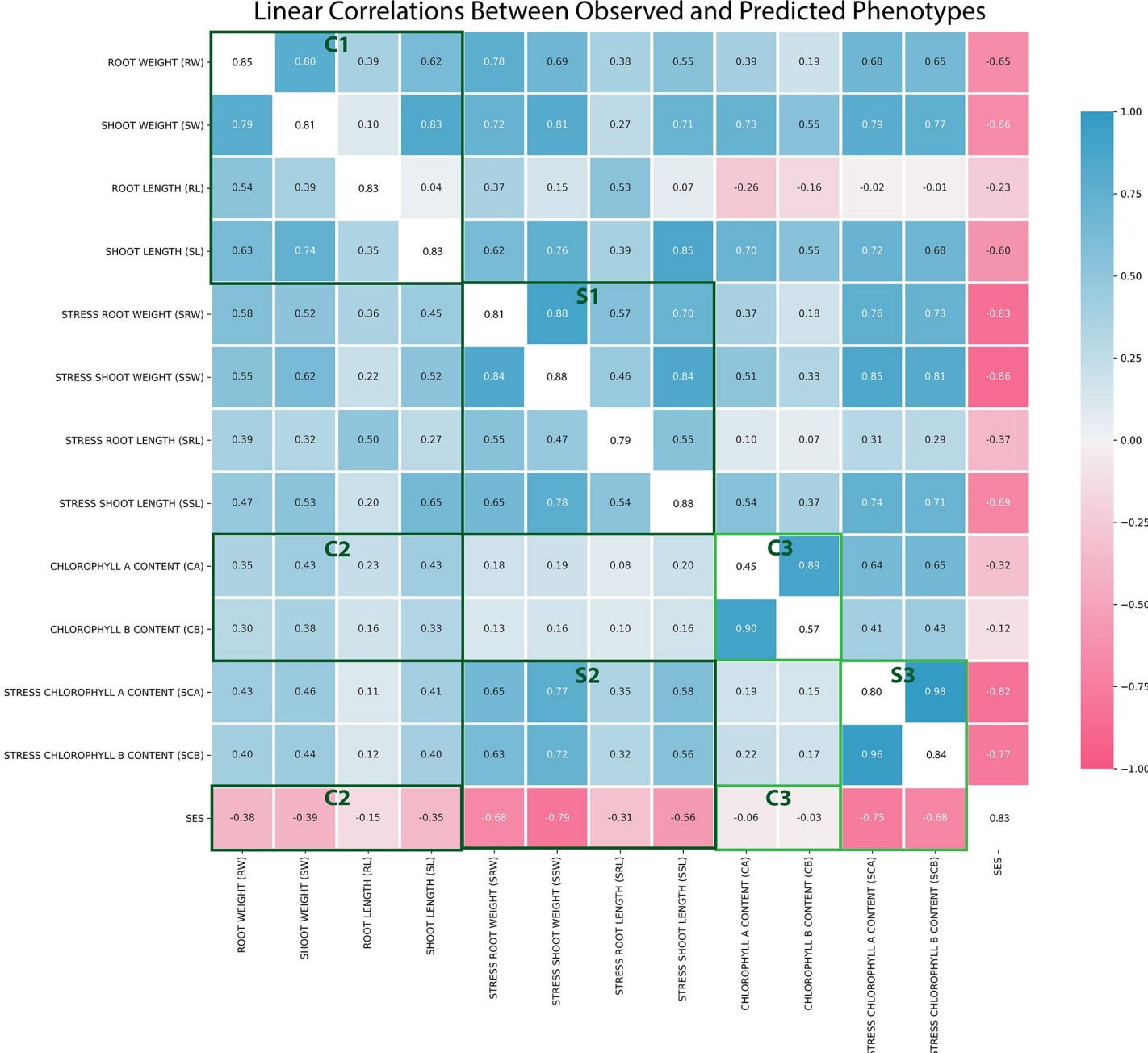

**Fig 3. Linear correlations (r) between observed and predicted phenotypes.** The bottom triangle of the white diagonal shows correlations between observed phenotype values and the top part is comprised of correlations between values estimated from genomic prediction. The white diagonal is the correlation between observed and predicted values for each phenotype. In the highlighted rectangles, C stands for control condition and S stands for stress condition. C1 to S1 and C2 to S2 depicts a fortification of linear correlation between biomass traits and SES when salt stress is imposed. C3 to S3 show the same trend for chlorophyll traits. C2 to S2 and C3 to S3 further show that the enhancement of linear correlation brought about by salt stress applies to SES as well.

correlation coefficients between the observed phenotypes are shown in the bottom diagonal of the heatmap in Fig 3 and the correlation between the genotype effects for each phenotype calculated through genomic prediction occupies the upper diagonal of the heatmap. The correlation between the observed phenotypes and additive SNP effects are seen along the white diagonal of the map. All phenotype estimations made by genomic prediction can be found in

S4 Table. An extended figure showing correlation coefficients between observed and estimated values of all 33 basic and derived phenotypes can be viewed in S1 Fig. A matrix of p values for the correlations shown in S1 Fig is provided in S5 Table.

In Fig 3, we observe better trait correlations in salt stress conditions relative to control conditions. Rectangle C1 and S1 compare correlations between biomass traits in control and stress conditions, C2 and S2 compare correlations between chlorophyll, SES and biomass in control and stress conditions. C3 and S3 compare correlations between chlorophyll and SES in control and stress. The improved linear correlations between biomass, chlorophyll and SES traits imply that the encumbrance of salt stress on a plant prevents the disproportionate gain of biomass. Since plant biomass and chlorophyll content is markedly reduced under salt stress (S3 Table), we can conclude that salt stress does not affect all phenotypes for all genotypes at the same rate and the genetic advantages that any genotype has in terms of tissue growth (root/shoot) or chlorophyll accumulation will be compromised by the effect of abiotic stress at a greater magnitude than traits which have near-baseline values. On account of linear correlation, plant root length stands out to be the most independent phenotype, especially in control conditions.

We derived principal components from the biomass and chlorophyll phenotypes under salt stress to compare the salt tolerability of our accessions. Since the composite SES score is a visual score broadly based on stunting of growth, discoloration of leaves and loss of plant vigor, we expect the pattern of SES scores to mimic the major principal component derived from the stress phenotypes. The first principal component could explain 69.6% variation in the data and has been plotted from the top in Fig 4 in the order of ascending SES score. From the figure, we observe a visible correspondence between SES score and the related stress condition phenotypes. In terms of SES score, the best performing varieties are IRIS_313–10973, IRIS_313–11224 and IRIS_313–8244 and the most sensitive varieties are IRIS_313–11063, IRIS_313–11064, IRIS_313–11400, IRIS_313–12055 and IRIS_313–8641. Based on the axis of maximum variation, the most tolerant accessions are IRIS_313–11116, IRIS_313–11217 and IRIS_313–9594 and the least tolerant accessions are IRIS_313–12055, IRIS_313–10606 IRIS_313–10602 and IRIS_313–8641 (S3 Table).

Density plots for all processed and derived phenotypes with subgroup specific curves for 82 Indica and 78 Aus varieties are tiled in S2 Fig. The biomass related traits we measured mostly followed a normal distribution with the exception of plant shoot length in both control and stress conditions. The ion content and chlorophyll content traits displayed large skews with few extreme values in both poles. Out of the ion content phenotypes, only shoot sodium content in stress condition exhibited a gaussian distribution (S2 Fig). Filtering out few extreme values mended the majority of the distributions to fit a normal curve. S6 Table lists the Shapiro-Wilk statistics for all phenotypes prior to filtering and the number of data points that needed to be omitted to improve the distribution. We did not use the filtered data in the primary association testing as it would compromise statistical power and risk spurious associations. Instead, data was transformed using an appropriate warping function and the changes in the distributions are shown in S3 Fig. Scatter plots of residuals from genomic estimates of phenotypes calculated before and after transformation can be seen in S4 Fig. The transformation method improved the centered clustering of residuals around zero and their uniformity in dispersion.

## 3.3 Genome-wide association studies

The BLINK algorithm [27] implemented in the GAPIT software package [28] is a recent innovation for powerful statistical association testing at considerably lower computational cost. BLINK achieves enhanced computational time by employing a linkage disequilibrium (LD)

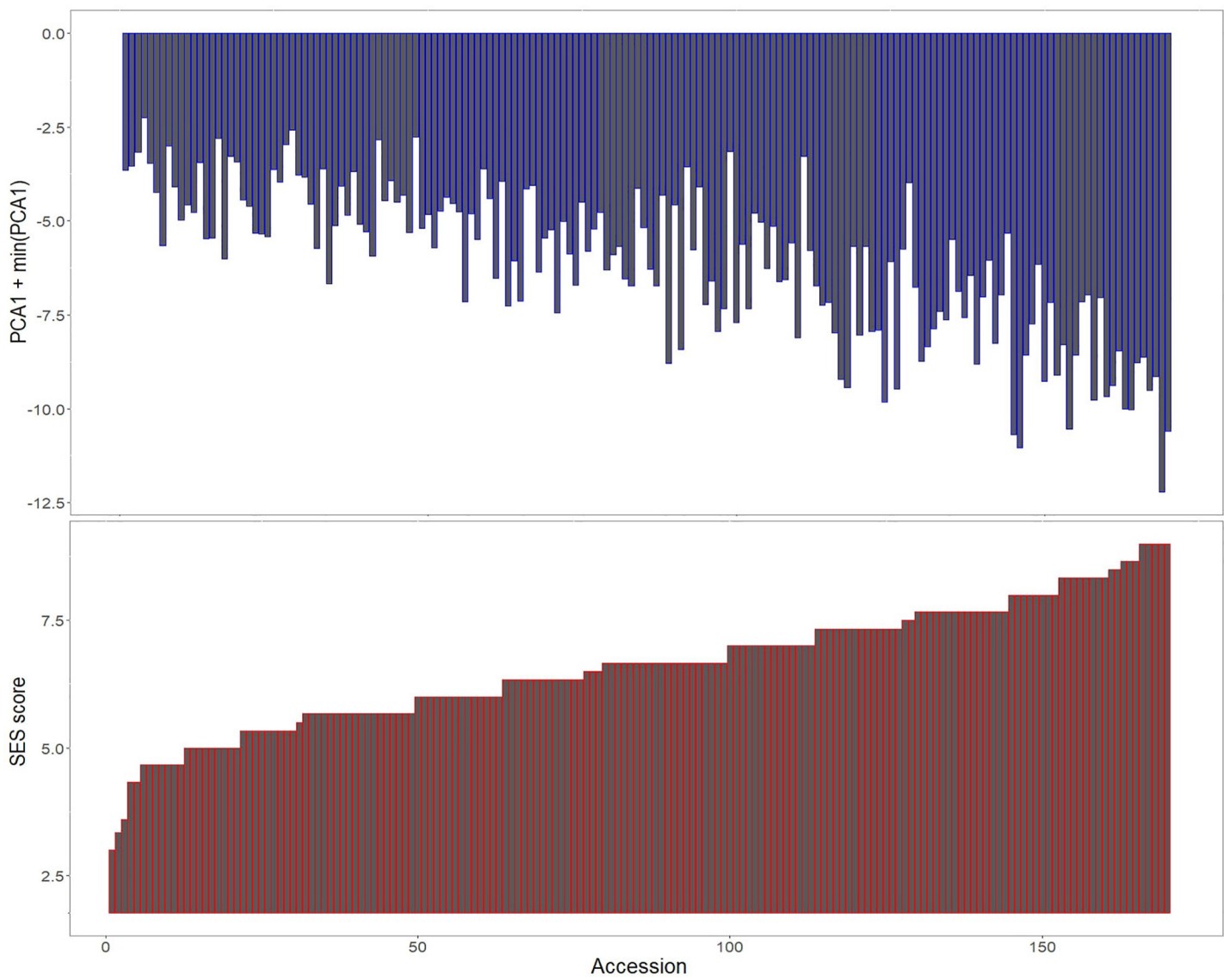

**Fig 4. Complementarity between SES scores and principal component 1 derived from biomass and chlorophyll content in stress conditions.** The principal component along the axis of maximum variation could explain 69.6% of the total variation. Metrics derived from simple numerical phenotypes could substitute for SES scores when a more objective index of salt injury is desired.

based iterative marker pruning strategy that fits two fixed effect models with a filtering step involving the second model. The algorithm is discussed in detail in the methods section. We also fit a more commonly used compressed mixed linear model (CMLM) to our phenotypes with optimum compression level to compare and confirm significant associations. Three independent studies on the whole population and Aus and Indica subpopulations were carried out respectively. 33 individual Manhattan and QQ plots each from three independent sets association tests are stored in the git repository under the 'GWAS' folder name. Table 2 lists 13 total QTLs identified and grouped into biomass or salt tolerance categories on the basis of their significant association to multiple traits belonging to the mentioned trait class. The first identified QTL, qCDP1.1, was observed in plant shoot length measured at stress conditions in the Indica subpopulation (Fig 5A). A suggestive peak on the same loci is found in the Manhattan plot for SES in the same subpopulation (S5 Fig). Two other associations noted for biomass traits were

**Table 2. Identified major QTLs.**

| Locus name | Trait class | Chromosome | Lead SNP position | P value | Traits where observed | Traits previously linked to by GWAS |
|---|---|---|---|---|---|---|
| qCDP1.1 | Biomass | 1 | 38423605 | 7.22E-07 | Stress shoot length, SES | Shoot length, shoot to root ratio and potassium concentration in salt stress [Leon et al. 2016]; root weight under phosphate starvation [Mai et al. 2020]; shoot weight, plant weight [To et al 2019]; shoot length, shoot length under zinc stress [Zhang et al. 2016]; shoot height in control and under aluminium stress [Tao et al. 2018]; root length under salt treatment, relative dry shoot weight, relative biomass [Zhang et al. 2020] |
| qCDP3.1 | Salt tolerance | 3 | 1528621 | 2.65E-08 | Shoot potassium | |
| qCDP5.1 | Salt tolerance | 5 | 17632907 | 2.26E-07 | Lost shoot length, lost shoot weight | Root length under phosphate deprivation [Mai et al. 2020]; total dry weight under salt stress [Yu et al. 2017] |
| qCDP6.1 | Salt tolerance | 6 | 20760852 | 2.23E-06 | Stress shoot length, stress root thickness | Salt injury score, plant dry weight under salt stress [Leon et al. 2016] |
| qCDP7.1 | Salt tolerance | 7 | 21913935 | 1.05E-06 | Stress root thickness, stress shoot thickness | Seed imbibition under salt stress [Cui et al. 2018]; relative maximum leaf length in salt stress [Frouin et al. 2018]; biomass, panicle dry weight, grain yield [Diop et al. 2020] |
| qCDP8.1 | Salt tolerance | 8 | 27323749 | 2.47E-06 | Lost shoot weight | Root length in salt stress[Leon et al. 2016]; spikely sterility [Dingkuhn et al. 2017]; resistance to bacterial blight [Delorean et al. 2016], potassium content and sodium-potassium ratio in salt stress [Frouin et al. 2018] |
| qCDP9.1 | Salt tolerance | 9 | 9438999 | 7.54E-06 | Lost shoot length | Low temperature seedling survivability [Schläppi et al. 2017] |
| qCDP9.2 | Salt tolerance | 9 | 11455731 | 1.91E-06 | Shoot potassium, chlorophyll A, chlorophyll B, lost root length, shoot sodium content | Seed germination rate under salt stress [Cui et al. 2018] |
| qCDP9.3 | Salt tolerance | 9 | 20273741 | 1.46E-06 | Stress shoot thickness, stress shoot weight, stress chlorophyll A | |
| qCDP10.1 | Salt tolerance | 10 | 18641874 | 1.07E-06 | Stress root potassium, root weight | SSI score, green leaves, shoot biomass, root biomass in salt stress [Rohila et al. 2019] |
| qCDP12.1 | Biomass | 12 | 2854998 | 6.51E-06 | Shoot thickness | Shoot weight to length ratio [To et al. 2019] |
| qCDP12.2 | Salt tolerance | 12 | 16595682 | 5.50E-06 | Stress root potassium, stress root sodium | |
| qCDP12.3 | Biomass | 12 | 25769505 | 7.57E-06 | Stress root length | Shoot length in salt stress [Leon et al. 2016]; low temperature seedling survivability and survival [Schläppi et al. 2017] |

qCDP12.1 and qCDP12.3. Like qCDP1.1, qCDP12.3 can also be found associated with stress tolerance traits (S5 Fig). qCDP3.1 relates to potassium content in plant shoots (Fig 6A). Because of the central role of potassium and other cations in salinity tolerance, we observe qCDP3.1 as a stress tolerance trait. Four more signals, qCDP5.1, qCDP6.1, qCDP8.1 and qCDP9.1, relating to the effect of salinity on plant shoots shown in Fig 6B and 6C were discovered. qCDP7.1, qCDP9.2 and qCDP9.3 concerning root and biomass traits are observed in Fig 7. Fig 8 shows that apart from qCDP3.1, two more loci, qCDP10.1 and qCDP12.2 are associated with ion content phenotypes in stress conditions. 10 out of 13 QTLs in our findings were previously discovered by GWAS studies with similar phenotype connotations (Table 2). An extended version of Table 2 with peak marker locations, original phenotype names and references [10, 11, 31–44] can be found in S7 Table.

## 3.4 QTL mapping

The extent of linkage disequilibrium in rice has been found consistently in the range of 100kb to 200kb [44], although some authors suggest that it could extend over 500kb for some

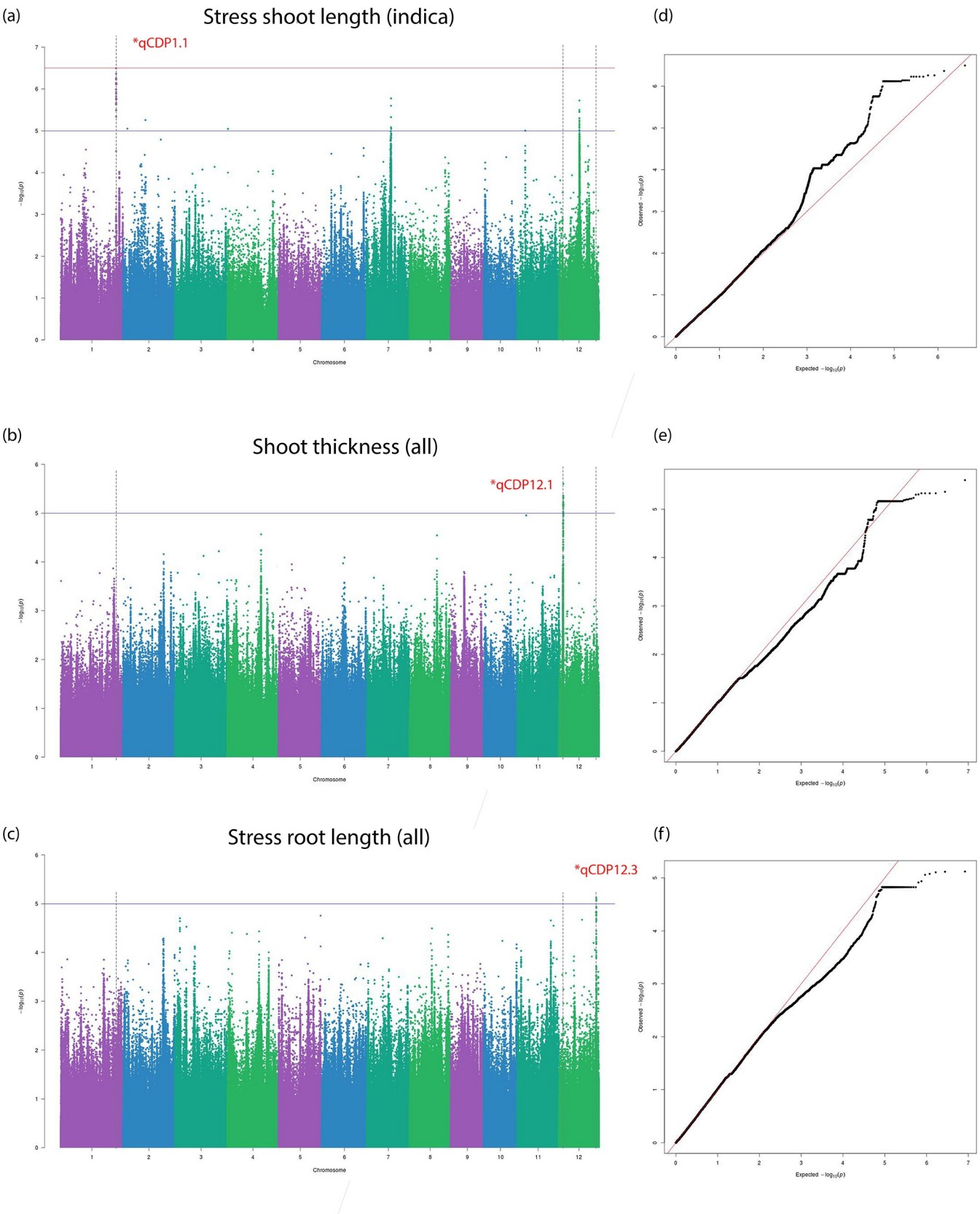

**Fig 5. Manhattan and Q-Q plots for qCDP1.1, qCDP12.1 and qCDP12.3.** Horizontal red and blue lines show the significant and suggestive thresholds respectively. Vertical black dotted lines show the locations of the three QTLs in the three Manhattan plots.

subpopulations [45]. In our experiments, LD decay to half maximal value of $r^2$ was observed at a distance of ~70kb in the whole population, ~100kb in the Aus subpopulation and ~50kb in the Indica subpopulation (S6 Fig). Based on these assumptions, we defined the range of our QTLs to be from 100kb upstream of the peak SNP up to 100kb downstream for association mapping. 313 markers showing significant and suggestive associations within the sequence range of known rice genes are described in S8 Table with metadata from the RAP-DB database [13]. All significant and suggestive markers from validation tests using the CMLM model can be found in S9 Table. We reviewed the annotations for genes situated in the vicinity of these significant associations for the elucidation of functional roles in regards to the implied phenotypes. From these annotations, 52 genes with notable functional roles corresponding to their associated phenotypes are included in the supplementary notes.

One significant marker from qCDP3.1 associated with shoot potassium content could be mapped to OsGSL10 (MSU ID: LOC_Os03g03610), a metabolic enzyme having 2-fold higher expression in shoots than any other tissue. In the same QTL, a cation antiporter associated to shoot potassium content, LOC_Os03g03590, could be mapped by a highly significant marker. This gene is also ubiquitously expressed with significantly higher expression in leaves and shoots. Based on its peptide sequence, the transporter protein is annotated to be localized in the chloroplast (Uniprot ID: Q10SB9). Within the same locus, two notable genes: OsCP29 (MSU ID: LOC_Os07g37240) and OsPetC (MSU ID: LOC_Os07g37030), are also localized in the chloroplast. Both these genes are expressed all throughout the plant in markedly higher expression levels and have almost 10-fold higher expression in leaf tissues. The presence of a chloroplast localized cation transporter adjacent to two particularly important chloroplast related genes in a gene locus identified through shoot potassium content suggests a role of this transporter protein and chloroplasts in the regulation of potassium ion content in plants. From the MSU7 brief loci annotations, genomic copies of the OsNAC6 gene (MSU ID: LOC_Os01g66120) which is known to have a functional role in plant development and stress tolerance, could be mapped to 4 identified QTLs: qCDP1.1 (LOC_Os01g66120), qCDP3.1 (LOC_Os03g03540), qCDP12.1 (LOC_Os12g05990) and qCDP12.3 (LOC_Os12g41680). qCDP1.1, qCDP12.1 and qCDP12.3 are QTLs belonging to our biomass category (Table 2). This is good evidence to suggest that OsNAC6 might have a prominent role in embryogenesis, germination and biomass gain in rice. The gene had also been implicated in stress response and abiotic stress tolerance which holds to its presence in the salt stress related QTLs: qCDP1.1, qCDP3.1 and qCDP12.3.

## 4. Discussion

### 4.1 Biomass traits and chlorophyll content could be valuable indices for the screening of salinity tolerance

There lies significant ambiguity in the extrapolation of salt tolerance mechanisms in rice [46]. Gregorio et al. 1997 [19] popularized visual injury scoring for the screening of salinity tolerance. The application of SES for assessment of salinity tolerance has gathered some criticism since then [14]. We found the SES score to have meaningful correlations with all observed traits except for ion content (S1 Fig). Our ion content traits are among the least heritable phenotypes (Table 1 and S2 Table) and their distributions deviate the most from a normal gaussian distribution (S2 Fig). They are also the most variable across genotypes and among

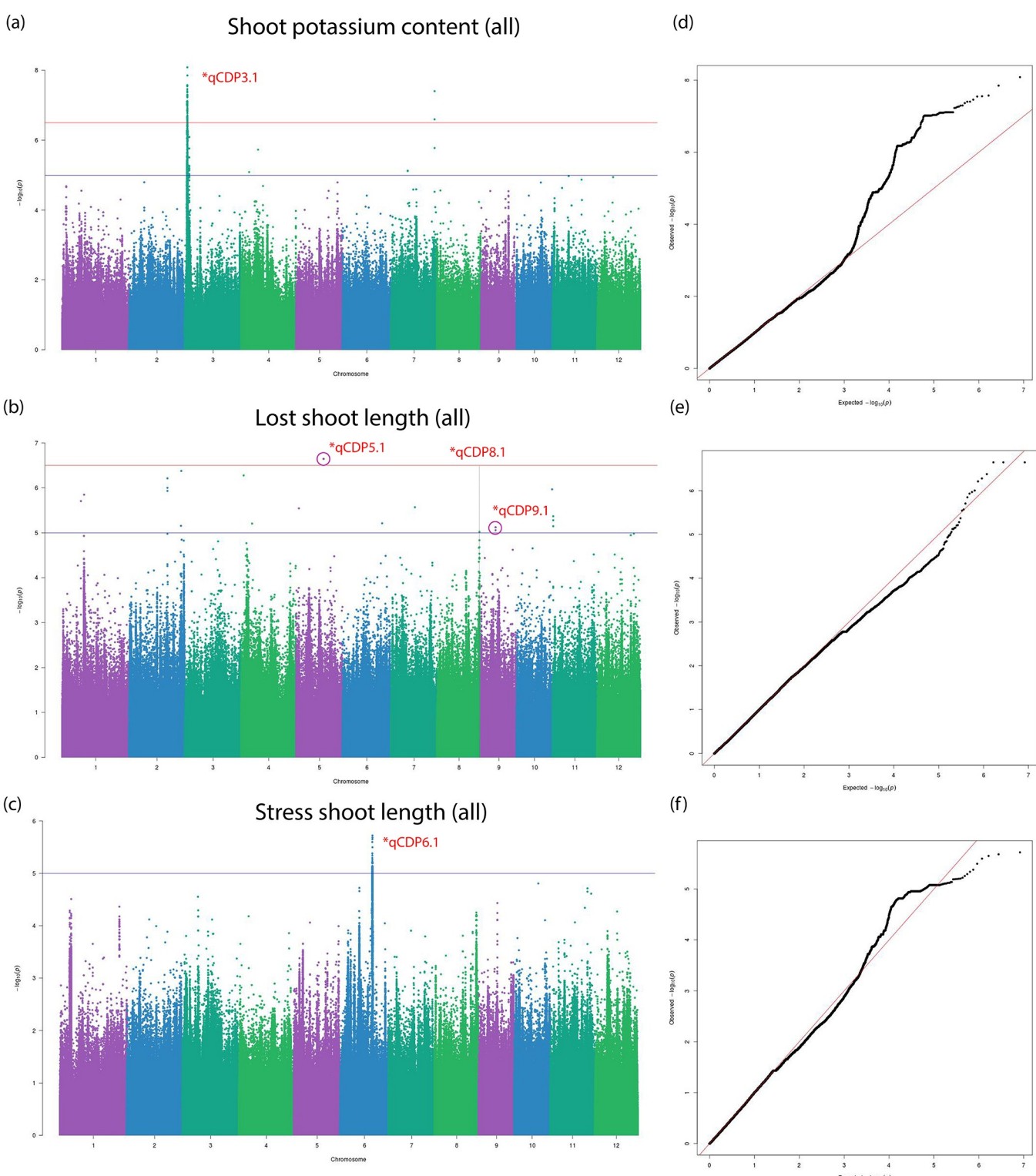

**Fig 6. Manhattan and Q-Q plots for qCDP3.1, qCDP5.1, qCDP6.1, qCDP8.1 and qCDP9.1.** Horizontal red and blue lines show the significant and suggestive thresholds respectively. Vertical black dotted lines show the locations of the five QTLs in the three Manhattan plots.

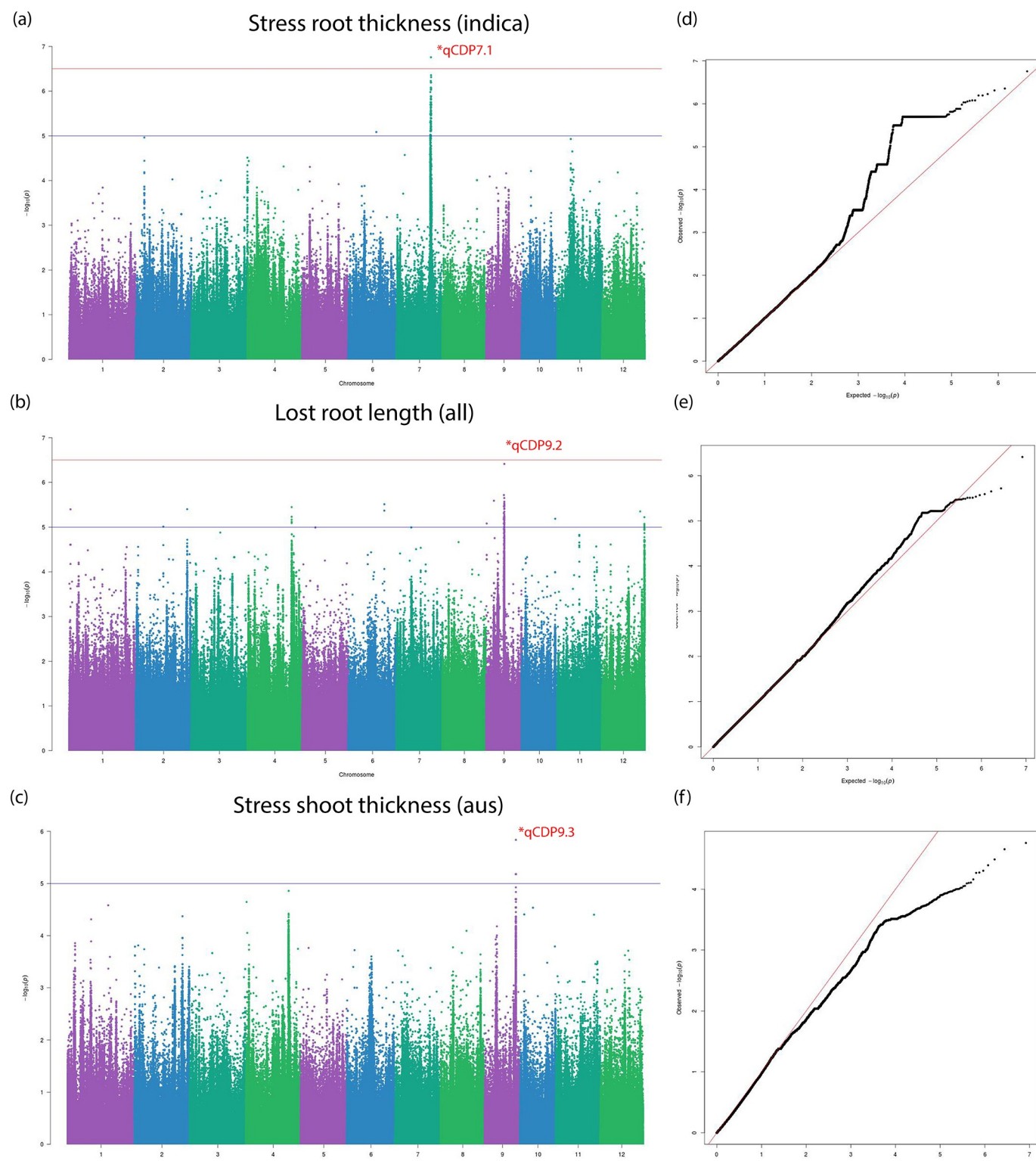

**Fig 7. Manhattan and Q-Q plots for qCDP7.1, qCDP9.1 and qCDP9.3.** Horizontal red and blue lines show the significant and suggestive thresholds respectively. Vertical black dotted lines show the locations of the three QTLs in the three Manhattan plots.

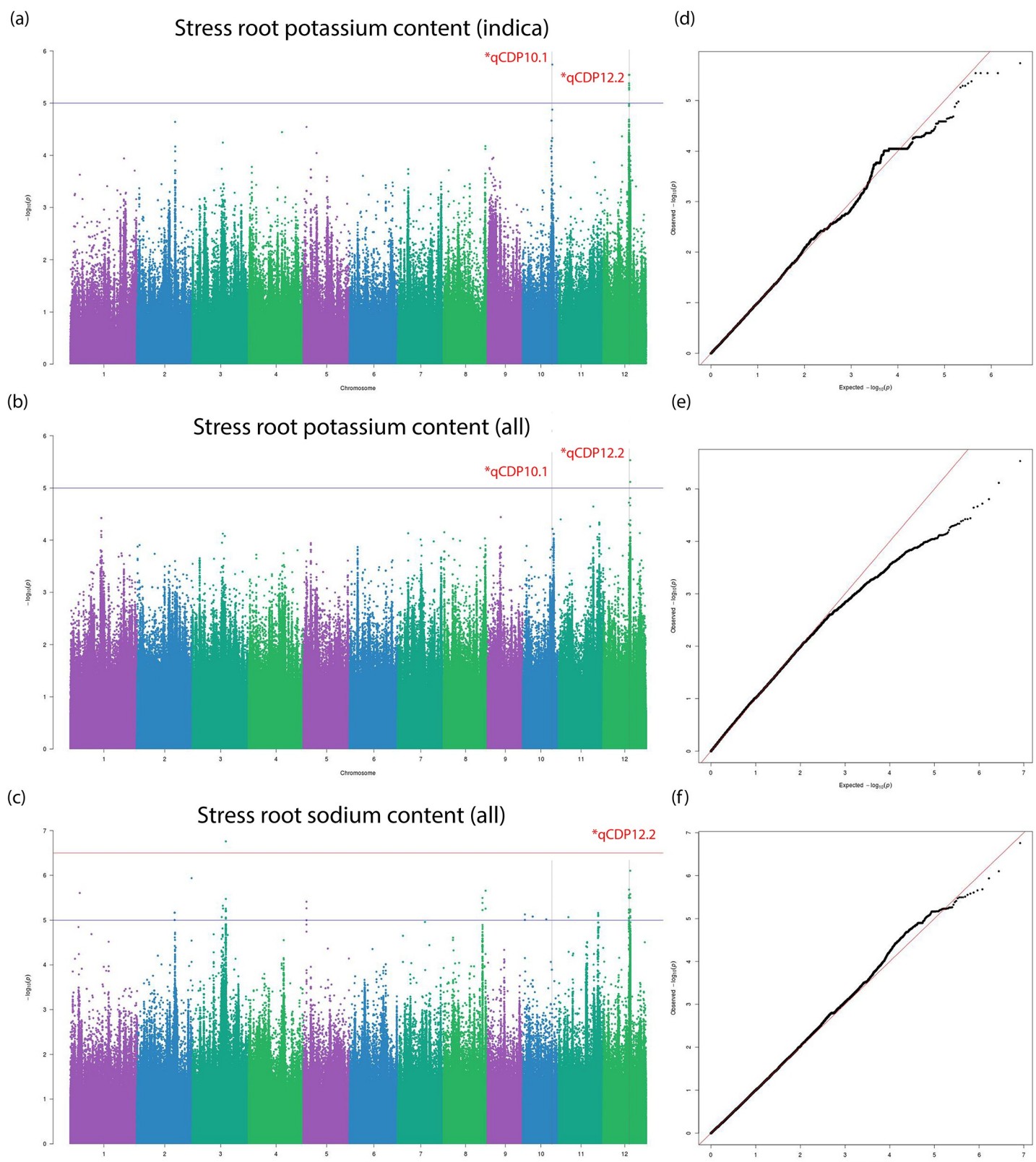

**Fig 8. Manhattan and Q-Q plots for qCDP10.1 and qCDP12.2.** Horizontal red and blue lines show the significant and suggestive thresholds respectively. Vertical black dotted lines show the locations of the two QTLs in the three Manhattan plots.

replicates (S3 Table), least applicable for genomic prediction (S4 Fig) and the least correlated between two test conditions and with the other phenotypes, including the composite SES score (S1 Fig). There are however, significant near-binary differences in tissue sodium content for control and stress counterparts and the magnitude of this change greatly outweighs the variance observed across genotypes (S3 Table). This proved that in spite of being the most significant physiological alteration with the most deleterious metabolic consequences, changes in sodium content for individual genotypes between control and stress conditions could not be an informative variable for prediction models. This lack of correlation and lack of reliable difference across genotypes was previously observed by Pires et al. 2015 [14]. Pires et al. also found that potassium content in roots and leaves were not significantly altered between control and stress group plants and thus, could not be a reliable predictor of salt tolerance in statistical models. There has already been ample criticism against the traditional application of the $K^+/Na^+$ ratio for evaluation of stress tolerance [14, 47] and our observations incurred further scrutiny to the conventional practice of evaluating salinity tolerance across different genotypes using tissue ion content and $K^+/Na^+$ ratios. For these reasons, we did not derive any further phenotypes from sodium and potassium content traits.

Although ion content is linearly independent to salt injury, very high correlation between sodium and potassium content in roots under salt stress is observed (S1 Fig). The lack of this correlation in shoots under stress conditions suggests that sodium exclusion from plant leaves might be critical for survival in salt stress. Chlorophyll A and B are not independent variables and although relatively less heritable, show high degree of correlation in both control and stress conditions and correlate strongly to salt injury. This was reported in Pires et al. 2015 and our findings confirm their observations. The relationship observed between biomass traits and salinity tolerance in our studies is noteworthy. It is always the combination of a number of salt tolerance strategies that a plant employs to cope with stress. Intuitively, salinity tolerance benchmarks are depicted by the ability of plants to accumulate and retain biomass and color. We observed that proportional losses of shoot and root biomass is instilled by salt stress. Whereas plant genotypes that had more chlorophyll in control conditions, did not retain more chlorophyll in stress conditions (Fig 3). Hence for chlorophyll, how much a plant has under normal conditions is not much relevant to how much a plant of the same genotype could maintain under salt stress conditions. How much biomass a plant of certain genotype retains in salt stress is however, linearly correlated to how much biomass it can put on in control conditions. Greater biomass in stress conditions correlates strongly and negatively to damage scores. It was also found by other researchers that the effect of salinity on biomass is more predominant in the shoots compared to roots [48]. It has previously been shown that response to salt stress in the tolerant landrace Pokkali takes place swiftly at the transcriptional level, whereas there is a delayed response in sensitive IR29 [49]. The homeostatic adjustments in tolerant plants allows them to continue growth, photosynthesize and ultimately set grains [50]. In light of these observations, and on the basis of GWAS results, we place much emphasis on biomass traits and chlorophyll content for the assessment of salinity tolerance in rice.

## 4.2 Some Bangladeshi landraces exhibit high degrees of salt tolerance

Two local endemic landraces, Kala Digha (IRIS_313–10973) and Kala Muna (IRIS_313–11115) are known to be highly salt tolerant. Similar degrees of tolerance were shown by Lal Digha (IRIS_313–11487) and Patnai 31–679 (IRIS_313–11224). The salinity tolerance of these accessions was in par with that of the tolerant landrace Pokkali (IRIS_313–8244) which is routinely cultivated in the coastal areas of Sri Lanka and Southern India. All of these accessions belong to the Ind2 subpopulation in the 3KRG database. The majority of our Ind2 plants are

moderately or highly salt tolerant whereas the sensitive plants mostly belong to the Aus sub-population (S3 Table). The varieties Chitraj (IRIS_313–11220), Kola Muchi (IRIS_313–10975) and Patnai 31–675 (IRIS_313–9049) stand out by having relatively high shoot biomass in spite of being highly salt sensitive. Kortik Kaika (IRIS_313–11116), Kal Shuli (IRIS_313–11113) and Pokkali (IRIS_313–8244) had the highest shoot biomass in stress conditions. The traditional varieties were overall more tolerant than the developed varieties. This is expected because developed varieties are generally bred and selected for yield traits rather than their wildtype characteristics that confer tolerance to environmental stress. In many places across the country where drought is common and along the southern coast where soil is saline, these local landra-ces are predominantly farmed as they perform better than the sensitive elite varieties.

## 4.3 Potential genes of interest were found in the identified QTLs

Studies on previously published rice gene expression data and functional annotations highlight a number of genes in the identified QTLs (S1 Text). Expression data of annotated genes within the QTLs taken from MSU7 are provided in S10 Table and processed RNA-seq data from the roots and shoots of a representative tolerant rice variety, Baldo and a sensitive variety, Vialone Nano adapted from Formentin et al. 2018 [51] are given in S11 Table. A heatmap of differen-tially expressed genes is shown in S7 Fig (see S1 Text for details). Effects of the natural poly-morphisms at the protein level can be found in S12 Table. Significant differences in phenotypes between the variant alleles are observed by ANOVA in S13 Table and by the Stu-dent's t-test in S14 Table.

Variant alleles of the chloroplast transporter gene discussed in the QTL mapping section (section 3.4), LOC_Os03g03590 found in qCDP3.1 did not exhibit significant differences in phenotype. OsPRX86 is a peroxidase precursor protein belonging to the QTL qCDP6.1. The locus has been previously found to be associated with salt injury scores in rice [10]. A polymor-phism causing a nonsense consequence in the gene was found to affect multiple salinity toler-ance traits, namely, SES score, shoot length under stress and normal conditions and shoot weight. Another unannotated gene, LOC_Os09g15389.2, belonging to another previously identified locus, qCDP9.1 [52], was observed to affect salt injury score, shoot weight and shoot length when the protein is truncated. OsPIL (MSU ID: LOC_Os12g41650.4) is a phytochrome interacting helix-loop-helix protein gene present in qCDP12.3 that has 5 to 10-fold greater expression levels in plant leaves, and is known to play a part in abiotic stress response [53]. Our observed SNP only affects transcript 4 out of 5 known transcripts of this gene. LOC_Os09g19160 is an uncharacterized serine/threonine kinase from qCDP9.2, that belongs to a cluster housing another serine/threonine kinase gene, LOC_Os09g19229. These kinases were found to affect root length in normal and stress conditions and also show significant dif-ferential expression under control and stress conditions. Similar kinases have been observed to play an important part in root development and nodulation [54]. An SNP variation in the 6-phosphogluconolactonase gene from qCDP8.1, LOC_Os08g43370.1 disrupted root length and shoot weight in stress condition. This gene also exhibits significant differential expression under control and stress conditions. These genes appear to be promising subjects for discrete functional studies to find any contributions they might have in abiotic stress tolerance.

## 5. Conclusion

We have mapped 13 QTLs for various traits including 3 novel QTLs. The novel QTLs qCDP3.1 and qCDP12.2 are associated with tissue ion content and qCDP9.3 is associated with plant biomass and chlorophyll content under stress conditions. The 10 other identified QTLs had previously been reported in literature against identical quantitative traits. When mapped

to the genome, these QTLs were found to harbor many genes implicated in abiotic stress response and tolerance. Supplementary analysis of gene expression and functional annotations assumes potential roles for a number of genes within the identified QTLs. A number of accessions from Bangladesh display high degree of salt tolerance. Some resilient and notable landraces include Kala Digha (IRIS_313–10973), Kala Muna (IRIS_313–11115) Lal Digha (IRIS_313–11487) and Patnai 31–679 (IRIS_313–11224). Although it is well established that biomass and chlorophyll related traits alongside tissue ion content aids the assessment of salinity tolerance, our studies show that the numerical values of plant biomass and chlorophyll content could manifest themselves into an objective index of salt injury unabated by human error.

GWAS studies are convenient protocols for novel genetic discoveries. They have become even more popular with the subsistence and growth of publicly available genomic data. The automation of genomic studies is more relevant today because of recent advancements in mechanical and drone-based phenotype imaging technologies. Public documentation of our experimental design on GitHub will assist future researchers and facilitate GWAS and other studies in plant genomics.

## Supporting information

**S1 Text. Gene expression and functional studies within the identified QTLs.**
(DOCX)

**S1 Fig. Extended correlation matrix of observed traits.**
(JPG)

**S2 Fig. Density plots for basic and derived phenotypes.**
(JPG)

**S3 Fig. Changes in phenotype distributions after applying an ideal warping function.**
(JPG)

**S4 Fig. Residual plots for basic and derived phenotypes.**
(JPG)

**S5 Fig. Collage of Manhattan plots for traits with biomass associations.**
(JPG)

**S6 Fig. Observed extent of linkage disequilibrium.**
(JPG)

**S7 Fig. Gene expression heatmaps from S1 Text.**
(JPG)

**S1 Table. Data on vegetation and population structure of 176 plants from the study.**
(XLSX)

**S2 Table. Heritability of traits calculated from genomic kinship and broad-sense heritability calculated from variance components.**
(XLSX)

**S3 Table. Observed phenotype means for each genotype.**
(XLSX)

**S4 Table. Phenotype values for genotypes estimated by genomic prediction.**
(XLSX)

**S5 Table. P values for entire correlation matrix in S1 Fig.**
(XLSX)

**S6 Table. Details on phenotype distributions before and after filtering.**
(XLSX)

**S7 Table. Details on QTLs from previous studies that were found to have similar connotations as our findings.**
(XLSX)

**S8 Table. All significant and suggestive marker associations that fall within the sequence range of known genes.**
(XLSX)

**S9 Table. All significant and suggestive marker association within known gene regions from validation tests by the CMLM model in GAPIT.**
(XLSX)

**S10 Table. Gene expression data for candidate genes.**
(XLSX)

**S11 Table. Gene expression data from leaves and roots of a sensitive and a tolerant variety.**
(XLSX)

**S12 Table. Effects of all 28188 possible substitution events within candidate gene regions.**
(XLSX)

**S13 Table. Results of One-way ANOVA carried out for all markers within candidate gene regions.**
(XLSX)

**S14 Table. Results of student's t test carried out for all markers within candidate gene regions.**
(XLSX)

## Acknowledgments

The authors would like to acknowledge Dr. Sabrina M. Elias from Independent University, Bangladesh and MU Sharif Shohan from the University of Dhaka for their valuable insights and guidance throughout the project. We acknowledge the Center for Bioinformatics Learning Advancement and Systematic Training (cBLAST) from the University of Dhaka and the University of Dhaka itself for enabling the necessary computational and experimental resources. We acknowledge Dr. Md. Sazzadur Rahman from Bangladesh Rice Research Institute (BRRI) for his assistance in multiplying seeds and preparing the germplasm. We also acknowledge the members of the Plant Biotechnology Lab at the University of Dhaka with special mention to Rabin Sarker and Raju Ahmed for help in handling plant material, screening for salinity tolerance and in enumerating and tabulating wet lab data. Finally, we acknowledge Anika Tahsin from the University of Dhaka for help in retrieving geolocational origins of the studied plant varieties and being a constant source of inspiration.

## Author Contributions

**Conceptualization:** Md Nafis Ul Alam, Zeba I. Seraj.

**Data curation:** Md Nafis Ul Alam, G. M. Nurnabi Azad Jewel, Tomalika Azim.

**Formal analysis:** Md Nafis Ul Alam, G. M. Nurnabi Azad Jewel, Tomalika Azim.

**Investigation:** Md Nafis Ul Alam.

**Methodology:** Md Nafis Ul Alam.

**Project administration:** Zeba I. Seraj.

**Software:** Md Nafis Ul Alam, G. M. Nurnabi Azad Jewel, Tomalika Azim.

**Supervision:** Zeba I. Seraj.

**Validation:** Md Nafis Ul Alam, Zeba I. Seraj.

**Visualization:** Md Nafis Ul Alam.

**Writing – original draft:** Md Nafis Ul Alam.

**Writing – review & editing:** Md Nafis Ul Alam, Zeba I. Seraj.

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
