## [Decision Letter · Decision Letter 0]

1 Jul 2021

PONE-D-21-14535

Comprehensive analysis and genome-wide association studies of biomass, chlorophyll, seed and salinity tolerance related traits in rice highlight genetic hotspots for crop improvement

PLOS ONE

Dear Dr. Seraj,

Thank you for submitting your manuscript to PLOS ONE. After careful consideration, we feel that it has merit but does not fully meet PLOS ONE’s publication criteria as it currently stands. Therefore, we invite you to submit a revised version of the manuscript that addresses the points raised during the review process.

    Specifically, both reviewers expressed concern regarding English language, redundancy, and some technical issues in this manuscript. Validation of few candidate genes is also suggested. I suggest to make a thorough revision with reorganization of contents without redundancy to make the manuscript reader-friendly.

We look forward to receiving your revised manuscript.

Kind regards,

Prasanta K. Subudhi, Ph.D.

Academic Editor

PLOS ONE

Additional Editor Comments:

Major revision

Journal Requirements:

2. We note that Figure 1 in your submission contain map images which may be copyrighted. All PLOS content is published under the Creative Commons Attribution License (CC BY 4.0), which means that the manuscript, images, and Supporting Information files will be freely available online, and any third party is permitted to access, download, copy, distribute, and use these materials in any way, even commercially, with proper attribution. For these reasons, we cannot publish previously copyrighted maps or satellite images created using proprietary data, such as Google software (Google Maps, Street View, and Earth). For more information, see our copyright guidelines: http://journals.plos.org/plosone/s/licenses-and-copyright.

2.1.    You may seek permission from the original copyright holder of Figure 1 to publish the content specifically under the CC BY 4.0 license. 

2.2.    If you are unable to obtain permission from the original copyright holder to publish these figures under the CC BY 4.0 license or if the copyright holder’s requirements are incompatible with the CC BY 4.0 license, please either i) remove the figure or ii) supply a replacement figure that complies with the CC BY 4.0 license. Please check copyright information on all replacement figures and update the figure caption with source information. If applicable, please specify in the figure caption text when a figure is similar but not identical to the original image and is therefore for illustrative purposes only.

Reviewers' comments:

Reviewer's Responses to Questions

**Comments to the Author**

1. Is the manuscript technically sound, and do the data support the conclusions?

Reviewer #1: Partly

Reviewer #2: Partly

2. Has the statistical analysis been performed appropriately and rigorously? 

Reviewer #1: Yes

Reviewer #2: I Don't Know

3. Have the authors made all data underlying the findings in their manuscript fully available?

Reviewer #1: Yes

Reviewer #2: Yes

4. Is the manuscript presented in an intelligible fashion and written in standard English?

Reviewer #1: No

Reviewer #2: No

5. Review Comments to the Author

Reviewer #1: The manuscript by Alam et al. titled “Comprehensive analysis and genome-wide association studies of biomass, chlorophyll, seed and salinity tolerance related traits in rice highlight genetic hotspots for crop improvement” describes a GWAS mapping analysis of 15 phenotypes yielding 17 QTL and list of 21 associated candidate genes using 176 mostly land race accessions from Bangladesh.

The general information provided by this manuscript and the QTL and candidate gene list are potentially useful for crop improvement of elite rice varieties challenged by abiotic stress such as high salinity. However, the manuscript is too long, needs to be streamlined to make it more reader friendly, and authors have not validated any of the 21 candidate genes. Below are specific comments that need to be addressed.

Line 58: “No recent GWAS study..” is too absolute as a statement; modify to “Few recent GWAS studies…”

Line 122: Change “Seed husks” to “Seed hulls”? Hulls is more commonly used than husks.

Lines 244-245 & Fig. 2a: what is the “blue” subpopulation? Is it aromatic?

Lines 285-286: what do you mean by “fortification of trait correlations in salt stress conditions relative to control conditions”? It is not clear how e.g. C1 compared to S1 in Fig. 3 allows you to come to this conclusion: what exactly is compered here? Fig. 3 is confusing in other ways: what do you mean by “Linear correlations between observed and predicted phenotypes”? How do you “predict” phenotypes? A genomic prediction method is mentioned on lines 149-141, but it is not clear how this SNP effect on phenotype is used for the Pearson’s correlation shown in Fig. 3.

Lines 262: since seed traits were only measured under control conditions, it is not clear how this data set is useful for abiotic stress tolerance improvement. To streamline the paper, please remove those data unless you can provide a good explanation in the manuscript for having them.

Lines 295-325: Some of the results section should be moved to the methods section or only briefly mentioned. The data transformation part to achieve normal distributions before GWAS analysis should be described in methods and Fig. 5 be included as a supplementary figure to streamline the manuscript.

Other result sections also contain quite a bit of methods descriptions that should be moved from Results to Methods.

Section 3.3., GWAS analysis, should be streamlined (again, do not describe methods here). The naming of QTL could be improved: instead of giving CDP consecutive numbers that seem to random, it would be useful to label a QTL on chromosome 1 as pCDP1-x etc (-x if there are more than one QTL on the same chromosome). Also, what does “CDP” stand for?

Fig. 10: heat map colors need to be quantified.

Lines 453-458: Should not refer to SNPs as “mutations”, b/c these are naturally or artificially selected variants

Section 4.1 (lines 479-494) does not contribute much to the discussion and could be removed or modified. What is the main message the authors what to convey?

Lines 612-613: Os09g19160 is mentioned twice but should be different genes.

Gene expression data were from publicly available RNAseq data only and not validated by the authors. Gene expression should be determined under both control and salt stress conditions for selecting the best candidate genes for crop improvement. Moreover, just focusing on genes with synonymous aa substitutions or frameshifts in some accessions might not be the best way to select candidate genes because they could be differentially expressed between stress tolerant and stress sensitive accessions. To improve the manuscript, validate a few candidate genes by qPCR in a selected stress tolerant and a selected stress sensitive accession.

This reviewer is also a bit confused with some aspects of the manuscript: wasn’t the main objective to identify salt tolerance QTL by looking at biomass and chlorophyll content under salt stress and perhaps looking at the relative reduction of both compared to control? So why the long discussion in section 4.3 about the chr. 3 chlorophyll QTL found under control conditions?

Lastly, the section 5 (Conclusion) could be improved: it is too long and should summarize the main points of the manuscript and should not read like an additional discussion.

Reviewer #2: The manuscript explains a GWAS study coupled with an extensive candidate gene analysis. The amount of effort made by authors is appreciable. However, there are serious concerns about the study and presentation. Some of my comments and suggestions are as below:

General comments:

1. Nowhere in result section authors have mentioned about novel QTLs identified in the study (they discussed about this in abstract & conclusion). To me this should be their key finding which is not explained properly.

2. Authors claimed a number of times about functional role of genes without experimental evidences which is not appropriate.

3. Results have not been explained, instead just the tables were referred in most of cases.

4. There is too much redundancy in write up, result section is full of material-methods,

5. Complex language is big problem in this MS. Authors should write simple language for science but they rather chosen very complex and idiomatic language. Some of the examples are: “handful of individuals”, “environment embellish immense diversity”, “scientists endeavor to elucidate”, “interactions with the surroundings underlie the quantitative assertion of these phenotypes”, “But such is customary to plant breeding experiments and many known and unknown factors are in play here.”, “contra-intuitive for practical purposes”, “an elegant-demonstration”, “Intuitively, salinity tolerance benchmarks are depicted by the ability of plants to accumulate and retain biomass and color.”, “It did not escape our notice”, “inspires us to suggest a novel role for the gene in seed dimension regulation”, “thrive in substandard land without any compromise in yield”, “we attempt to rapidly sieve these desirable attributes into our modern cultivars”, “These observations are however, not inconceivable.”, “firm belief” and many more.

Specific comments:

Line 58-59: “No recent GWAS study explores abiotic stress tolerance in relation to developmental and agronomic phenotypes in rice.” – there are plenty of GWAS studies on agronomic traits, presume that agronomic phenotypes relates to agronomic traits. What author means developmental phenotype is probably evolutionary basis but how does this MS linked to GWAS for the evolutionary traits? Didn’t find any evolutionary in-depth analysis in this MS except grouping which is very common.

Line 62 – 64: CRISPR is very much functional but with targeted gene insertion or knock-outs. Is there any example where QTL that carries multiple genes spanning very long DNA fragment substituted the alternate allele through CRISPR and worked? Or it is just hypothesis?

Line 83: What do you mean by ‘176 rice accessions ordered from IRRI seed bank’? Author can say got the accessions or received from IRRI gene bank.

Line 84-85: “BRRI (Bangladesh Rice Research Institute) fields where the day/night temperature was 32°/28°C”. Is it possible to have the same temperature in field throughout cropping season?

Line 97-98: Screening for salinity tolerance at seedling stage was carried out as per Amin et al (2012). This article is not an open access article hence could not go through the details but as mentioned in M & M, germinated seeds grown for 14 days the salt stress applied in 5 increments from 4 to 12 dS/m. Then phenotypic measurements were made after 16 days from 4 dS/m treatment (30 days). SES scores and other parameters were taken as described by Gregorio et al 1997. The screening as per Amin et al and measurement as per Gregorio et al are two different things and will produce two different scenarios. Gregorio used to treat the plant when they are really at seedling and single tiller stage (first treatment of 6 dS/m at 6th days from germination and raised to 12 dS/m on 9th day). Once the seedling becomes big enough (started multiple tillering), the response to salinity becomes different, so not sure why authors used old seedlings to screen when they already start attaining the tolerance to salinity.

Line 117 and 118: What are the details of the formulae to measure the chlorophyll? Volume in ml or l? Weight (fresh or dry) in, mg or g ? Absorbance etc. Explain each formula.

Line 153: 2,930,739 missing ‘genotypes’? – it is typo

Line 159-161: Why MAF was chosen to 10% for subpopulations : Should be explained.

Line 197-199: Authors have discussed a lot about the FDR (through Bonferroni), however, more important is the effect of population structure on associations. The way authors have explained above about the model for testing marker effects (in two equations), authors can explain OR simplify the explanation about population structure correction.

Results: For presenting the results, authors have just referred the tables/ figures and in text there is too much redundancy with material-methods. Result section needs to be re-organized in way it befits for a research paper, explaining the crystal-clear results ONLY. Say for example line 226 to almost 239 (except 236 and 237), is superfluous (“largest delta on Earth with bountiful rivers draining into the northeastern part of the Indian Ocean by way of the Bay of Bengal”) or part of M & M. There are plenty of instances where information fit for ‘M & M’ are written in ‘Results’ like 245-247, 254-257, 259-265, 327-337 and so on).

Line 274 – 276: Why it is so, please explain: “Narrow-sense heritability could not be calculated from additive marker effects because the grand mean of the phenotypes could explain more variance than the mixed.solve prediction model”

Line 306: What is bulky seed in term of their shape and size? Heard for the first time.

Line 361: Authors need to explain simply that what novel QTLs/ association they found and what was their effect. Their relation with candidate genes can be explained in next section. Could not find the information about the novel QTL in this section (QTL mapping) which could be the most important basis for acceptance of article for publication.

Line 377: What is trait like seed height? I have heard about seed length but not seed height? Are they same? If they are same, then why mentioned as “We previously found seed length to have

no correlation with seed height” (line 607-698).

Line 468-475: This can go as the legend or the footnote of the table.

Line 481-487: “Individual differences in plants of identical genotype are disturbingly disorderly. We found that variance within genotypes in our growth experiments were distinctly large for most traits other than seed properties. But such is customary to plant breeding experiments and many known and unknown factors are in play here. The question here is not how we can remove these confounding variables, since it is not only impossible but also contra-intuitive for practical purposes, but rather how we can derive informative features from observable phenotypes.”. What authors want to convey is not clear. The message should be simple and clear. ‘Individual differences in plants of identical genotype’ : It is not new when you are working with landraces. I am fail to understand the message out of this.

Line 493 – 494: This is well establish fact, what is new in this?

Line 574: Heading indicate the table number (“More candidate genes in table 4”)? Can’t be some better heading?

Line 627-628: Concluding a gene/QTL/marker effect, should require concrete experimental results. Just annotation based on literature does not make proper sense. Yes, you can relate but concluding is different and should be based on sound demonstrated results. Same for the functional role for the protein (where are results ? – line 630-631).

6. PLOS authors have the option to publish the peer review history of their article (what does this mean?). If published, this will include your full peer review and any attached files.

Reviewer #1: No

Reviewer #2: No

---

## [Author Response · Author response to Decision Letter 0]

22 Jul 2021

We appreciate the concerns raised by the reviewers. Their scrutiny is by and large logical and reasonable. We have made extensive changes throughout the manuscript to address most of their comments and voiced our rebuttals for only a few out of all the points they have raised.

In this revision, we have made an effort to improve the language and organization of the manuscript and particularly refine the contents of the results section. We have incorporated gene expression data collected from a previous transcriptome analysis of a sensitive and a tolerant variety of rice where the plants were subjected to salt stress to assess differential expression [1]. All seed related data and results including 4 QTLs and all candidate genes and SNP effects from those QTLs have been removed with hopes of streamlining the manuscript to better fit a salinity stress tolerance related category.

We acknowledge that the weakest aspect of our experiments is the absence of thorough functional studies on the QTLs for validation at the gene and protein level. Statistical and GWAS studies have been carried out at length for a large number of phenotypes and this has led to a very long manuscript. There are many GWAS papers being published today that do not include functional studies [2-10]. This started out as a student project. As such the work not only had time constraints but was not budgeted for either. Therefore, we wish to avoid going into functional studies for this manuscript and rather build upon the current body of work in the near future by undertaking new projects. 

Response to reviewer #1

We are thankful that the reviewer finds merit in our work. We address their concerns in detail below.

Comment: Line 58: “No recent GWAS study..” is too absolute as a statement; modify to “Few recent GWAS studies…”

Response: Modified in line 55.

Comment: Change “Seed husks” to “Seed hulls”? Hulls is more commonly used than husks.

Response: Seed related traits and results were removed as per the reviewer’s suggestion.

Comment: Lines 244-245 & Fig. 2a: what is the “blue” subpopulation? Is it aromatic?

Response: Yes. We have labelled it in figure 2 to make it more explicit.

Comment: what do you mean by “fortification of trait correlations in salt stress conditions relative to control conditions”? It is not clear how e.g. C1 compared to S1 in Fig. 3 allows you to come to this conclusion: what exactly is compered here? Fig. 3 is confusing in other ways: what do you mean by “Linear correlations between observed and predicted phenotypes”? How do you “predict” phenotypes? A genomic prediction method is mentioned on lines 149-141, but it is not clear how this SNP effect on phenotype is used for the Pearson’s correlation shown in Fig. 3.

Response: Figure 3 was inspired by Figure 2 (a) of Crowell et al. 2015 [11]. A correlation matrix is essentially a triangle, since you have one square in the heatmap for every one-to-one comparison. The correlations between the means of observed phenotype values given in supplementary table 3 are seen in the lower triangle of the heatmap and the correlation coefficients between the estimated phenotype values given in supplementary table 4 are given in the upper triangle. The diagonal given in white is simply one-to-one comparisons between the observed value and the predicted value of a given phenotype. 

Phenotypes can be estimated from SNP and observed phenotype data using the mixed.solve() function. The function calculates a base value for each phenotype and then estimates a phenotype for a variant by adding up the marker effects of each SNP marker in that individual. Thus, the upper triangle of the figure is more representative of the correlations between the phenotypes when the difference in values between the variants (supplementary table 4) is entirely based on differences in their genotype. 

Fortification of trait correlation simply means that stressed plants have a more linear relationship (heatmap colors are deeper) between certain phenotypes (marked by green rectangles in figure 3) than plants under control conditions.

We have added the following lines (255-258) in the manuscript to make the exposition of this figure more reader-friendly: “Rectangle C1 and S1 compare correlations between biomass traits in control and stress conditions, C2 and S2 compare correlations between chlorophyll, SES and biomass in control and stress conditions. C3 and S3 compare correlations between chlorophyll and SES in control and stress.”

Comment: Lines 262: since seed traits were only measured under control conditions, it is not clear how this data set is useful for abiotic stress tolerance improvement. To streamline the paper, please remove those data unless you can provide a good explanation in the manuscript for having them.

Response: All data, results and discussions about seed traits have been omitted in this revised version.

Comment: Lines 295-325: Some of the results section should be moved to the methods section or only briefly mentioned. The data transformation part to achieve normal distributions before GWAS analysis should be described in methods and Fig. 5 be included as a supplementary figure to streamline the manuscript.

Response: Extensive changes to the results section has been made. Data distribution image was moved to supplementary as suggested. 

Comment: Other result sections also contain quite a bit of methods descriptions that should be moved from Results to Methods.

Response: We have omitted the redundant methodological details from the results.

Comment: Section 3.3., GWAS analysis, should be streamlined (again, do not describe methods here). The naming of QTL could be improved: instead of giving CDP consecutive numbers that seem to random, it would be useful to label a QTL on chromosome 1 as pCDP1-x etc (-x if there are more than one QTL on the same chromosome). Also, what does “CDP” stand for?

Response: Results were streamlined and naming convention for the QTLs were changed as per the author’s suggestion. CDP is simply the initials for the original project title: Comprehensive Data for Plants.

Comment: Fig. 10: heat map colors need to be quantified.

Response: The image (Figure 8 (a) now) was modified to add the color key.

Comment: Lines 453-458: Should not refer to SNPs as “mutations”, b/c these are naturally or artificially selected variants.

Response: The authors are embarrassed about this mishap and it has been corrected throughout.

Comment: Section 4.1 (lines 479-494) does not contribute much to the discussion and could be removed or modified. What is the main message the authors what to convey?

Response: The section has been removed entirely.

Comment: Lines 612-613: Os09g19160 is mentioned twice but should be different genes.

Response: Fixed.

Comment: Gene expression data were from publicly available RNAseq data only and not validated by the authors. Gene expression should be determined under both control and salt stress conditions for selecting the best candidate genes for crop improvement. Moreover, just focusing on genes with synonymous aa substitutions or frameshifts in some accessions might not be the best way to select candidate genes because they could be differentially expressed between stress tolerant and stress sensitive accessions. To improve the manuscript, validate a few candidate genes by qPCR in a selected stress tolerant and a selected stress sensitive accession.

Response: We have included gene expression data from a previous transcriptome analysis [1] of a sensitive and a tolerant variety of rice. Formentin et al. 2018 [1] had carried out deep RNA-seq studies in both control and stress conditions. Their data was acquired from supplementary table 2 of their study and further processed to facilitate necessary comparisons between our candidate genes in control vs stress conditions and in a sensitive and a tolerant check.

Comment: This reviewer is also a bit confused with some aspects of the manuscript: wasn’t the main objective to identify salt tolerance QTL by looking at biomass and chlorophyll content under salt stress and perhaps looking at the relative reduction of both compared to control? So why the long discussion in section 4.3 about the chr. 3 chlorophyll QTL found under control conditions?

Response: The section has been removed.

Comment: Lastly, the section 5 (Conclusion) could be improved: it is too long and should summarize the main points of the manuscript and should not read like an additional discussion.

Response: The conclusions section has been edited to only include the bare essentials.

Response to reviewer #2:

The authors are grateful for the critical comments and expert evaluation of the reviewer. We have made an effort to address all concerns raised and improve the manuscript in light of their suggestions.

Comment: 1. Nowhere in result section authors have mentioned about novel QTLs identified in the study (they discussed about this in abstract & conclusion). To me this should be their key finding which is not explained properly.

Response: We have made a lot of changes to the results section in this revision and hope that it does a better job of highlighting the key findings of our study. The GWAS results in section 3.3 and table 2 lead directly to the genes mapped to the QTLs in section 3.4 and table 3. Section 3.5 and 3.6 explore additional analysis which are brought together in tables 3 and 4. The QTLs have been discussed in the text in section 3.3 lines 295-308. There is further discussion about the genes within these QTLs in lines 323-344, lines 367-374, and also in section 4.2.

Comment: 2. Authors claimed a number of times about functional role of genes without experimental evidences which is not appropriate.

Response: The authors completely agree that is extremely inappropriate to make any claims without concrete experimental evidence. We strive to not make any unsubstantiated claims about our findings. We have noted the gene roles as ‘potential’ gene roles in the introduction, result and conclusions. 

We acknowledge the absence of functional validation to be the main shortcoming of our manuscript, but as we have cited at the very beginning of this response letter, there have been many GWAS studies that were considered complete and suitable for publication without specific functional studies on the QTLs or genes.

Comment: 3. Results have not been explained, instead just the tables were referred in most of cases.

Response: Extensive edits and rewrites have been carried out on the results section to correct the flaws of the previous version. We have labored to include an exposition for every table and figure that has been cited in the text. For example, figure 1 and 2 stem directly from supplementary table 1, lines 236-238 observe supplementary table 2, data from supplementary tables 3, 4 and 5 give rise to figure 3, and so on.

Comment: 4. There is too much redundancy in write up, result section is full of material-methods.

Response: Major changes have been made to fix these issues.

Comment: 5. 5. Complex language is big problem in this MS. Authors should write simple language for science but they rather chosen very complex and idiomatic language. Some of the examples are: “handful of individuals”, “environment embellish immense diversity”, “scientists endeavor to elucidate”, “interactions with the surroundings underlie the quantitative assertion of these phenotypes”, “But such is customary to plant breeding experiments and many known and unknown factors are in play here.”, “contra-intuitive for practical purposes”, “an elegant-demonstration”, “Intuitively, salinity tolerance benchmarks are depicted by the ability of plants to accumulate and retain biomass and color.”, “It did not escape our notice”, “inspires us to suggest a novel role for the gene in seed dimension regulation”, “thrive in substandard land without any compromise in yield”, “we attempt to rapidly sieve these desirable attributes into our modern cultivars”, “These observations are however, not inconceivable.”, “firm belief” and many more.

Response: We feel that the reviewer is perhaps a little too scrupulous here. The phrase “It did not escape our notice” has been adapted from the last paragraph of James Watson and Francis Crick’s landmark paper on the discovery of the double helix where they hint at the implications of their discovery in explaining the molecular basis of Mendelian inheritance. 

Nonetheless, many of these quoted sections have been removed and with many others, we have made changes to accommodate the reviewer’s preferences as follows:

Line 229: “handful of individuals” was changed to “few individuals”.

“environment embellish immense diversity” was removed.

Line 48 “scientists endeavor to elucidate” was changed to “it has become possible to elucidate”.

Line 49: “interactions with the surroundings underlie the quantitative assertion of these phenotypes” was left unchanged.

“But such is customary to plant breeding experiments and many known and unknown factors are in play here.”, section was removed.

“contra-intuitive for practical purposes”, section was removed.

Line 432: “an elegant-demonstration”, the authors feel that it truly is an elegant demonstration, so left unchanged.

Line 438: “Intuitively, salinity tolerance benchmarks are depicted by the ability of plants to accumulate and retain biomass and color.”, the authors feel that it conveys an important message and so left unchanged.

“It did not escape our notice”, section was removed.

“inspires us to suggest a novel role for the gene in seed dimension regulation”, section was removed.

“thrive in substandard land without any compromise in yield”, was removed.

“we attempt to rapidly sieve these desirable attributes into our modern cultivars”, was removed.

“These observations are however, not inconceivable.”, was removed.

Line 500: “firm belief”, left unchanged because the authors feel that an individual’s beliefs must always be firm and infallible.

Specific comments by reviewer #2

Comment: Line 58-59: “No recent GWAS study explores abiotic stress tolerance in relation to developmental and agronomic phenotypes in rice.” – there are plenty of GWAS studies on agronomic traits, presume that agronomic phenotypes relates to agronomic traits. What author means developmental phenotype is probably evolutionary basis but how does this MS linked to GWAS for the evolutionary traits? Didn’t find any evolutionary in-depth analysis in this MS except grouping which is very common.

Response: The authors have yet to come across any GWAS study that explores biomass, ion content, chlorophyll and seed traits in conjunction and this had been the basis of this statement. But as per the suggestions of reviewer #1 (comment 1) and reviewer #2 here, we have modified the sentence to “few recent GWAS studies…” in line 55.

Comment: Line 62 – 64: CRISPR is very much functional but with targeted gene insertion or knock-outs. Is there any example where QTL that carries multiple genes spanning very long DNA fragment substituted the alternate allele through CRISPR and worked? Or it is just hypothesis?

Response: We completely agree with the reviewer. We merely suggest that the genes within these QTLs are amenable to CRISPR-Cas9 editing. We have changed ‘salient QTLs’ to ‘salient features’ in line 59 to clarify the statement.

Comment: Line 83: What do you mean by ‘176 rice accessions ordered from IRRI seed bank’? Author can say got the accessions or received from IRRI gene bank.

Response: ‘ordered’ was changed to ‘received’ in line 79.

Comment: Line 84-85: “BRRI (Bangladesh Rice Research Institute) fields where the day/night temperature was 32°/28°C”. Is it possible to have the same temperature in field throughout cropping season?

Response: These numbers denote the average ambient temperature of the cropping season. We have edited the section to include the months of the cropping season. Quoting lines 80 to 82: “Seeds were multiplied at BRRI (Bangladesh Rice Research Institute) fields in the months of March and April when the average day/night temperature was 32°/28°C with an average humidity of 72%.”

Comment: Line 97-98: Screening for salinity tolerance at seedling stage was carried out as per Amin et al (2012). This article is not an open access article hence could not go 

through the details but as mentioned in M & M, germinated seeds grown for 14 days the salt stress applied in 5 increments from 4 to 12 dS/m. Then phenotypic measurements were made after 16 days from 4 dS/m treatment (30 days). SES scores and other parameters were taken as described by Gregorio et al 1997. The screening as per Amin et al and measurement as per Gregorio et al are two different things and will produce two different scenarios. Gregorio used to treat the plant when they are really at seedling and single tiller stage (first treatment of 6 dS/m at 6th days from germination and raised to 12 dS/m on 9th day). Once the seedling becomes big enough (started multiple tillering), the response to salinity becomes different, so not sure why authors used old seedlings to screen when they already start attaining the tolerance to salinity.

Response: The authors have tried to convey that the screening protocol in its entirety mimics the works of Amin et al. 2012 [12]. Gregorio et al. 1997 [13] is only referenced in relation to the criteria for salt injury scoring in terms of SES. Rice plants are still seedlings at 14-30 days of age and at the time of harvest for our analysis, they had not started producing any tillers at all. These methods have been devised and followed as appropriate for screening of salinity tolerance at the seedling stage. Another point to note here is that our studies were in a net house, where the growth of the seedlings is dependent on the weather. Therefore we observe the size of the plant rather than the days. The variation in the days when the plants are deemed ready is usually between 12-16 days. The plants in this case were of different genotypes and the sizes varied between 6-8 inches, when the salt exposure regime was started. Earlier salt exposure would have been appropriate for an exclusive post-germination/early-seedling stage tolerance study.

Comment: Line 117 and 118: What are the details of the formulae to measure the chlorophyll? Volume in ml or l? Weight (fresh or dry) in, mg or g ? Absorbance etc. Explain each formula.

Response: The recommended explanations have been appended. Quoting lines 114 to 116: “Here A_663 and A_645 refer to the absorbance at 663nm and 645nm wavelengths respectively. V is volume in milliliters, W is the fresh weight of the leaf pieces in milligrams and chlorophyll content is expressed as milligrams per gram of fresh leaf tissue.”

Comment: Line 153: 2,930,739 missing ‘genotypes’? – it is typo.

Response: These refer to individual SNP marker genotypes not the whole genome/genotype of the individual. The authors affirm that this terminology is common in computational genomics to refer to the ‘genomic configuration’ of the SNP markers.

Comment: Line 159-161: Why MAF was chosen to 10% for subpopulations : Should be explained.

Response: The purpose of the minor allele filter is to filter out rare alleles that occur in very low frequency because the inclusion of these rare alleles in the study compromises statistical power and leads to spurious associations. MAF of 5% for 176 individuals means there has to be at least 9 instances of the minor allele, which is a suitable number for GWAS, but 5% of 82 or 78 is only ~4 which could yield false positive results with very low p-values. That is why a 10% MAF is more suitable for our subpopulation-based GWAS tests. We have added the following in lines 149 to 151 in the methods section to explain this briefly: “For GWAS, a minor allele frequency filter of 5% was applied to the whole population. Because of the smaller sample size, the minor allele frequency filter for the subpopulations was set at 10%.”

Comment: Line 197-199: Authors have discussed a lot about the FDR (through Bonferroni), however, more important is the effect of population structure on associations. The way authors have explained above about the model for testing marker effects (in two equations), authors can explain OR simplify the explanation about population structure correction.

Response: The authors agree that the BLINK algorithm can seem quite complex. But population structure correction is a relatively simple part of the model and is devoid of the complexity present in the exposition of the BLINK algorithm in section 2.7. A kinship matrix for the study sample and principal components from the genotype data are calculated and they are simply included in the final model as covariates after the BLINK algorithm has been executed. This has been explained in lines 184 to 187 of section 2.7: “To correct for population structure and cryptic relatedness, the kinship matrix K, calculated using GAPIT and the Q matrix having the first three principal components were fit in the first model as covariates. For the compressed mixed linear models fit for validation tests, the optimum compression level was selected and the same covariates were included.”

The authors feel that further explanation would needlessly inflate the methods section even further. Additionally, readers can always refer to the original article on the BLINK algorithm [14].

Comment: Results: For presenting the results, authors have just referred the tables/ figures and in text there is too much redundancy with material-methods. Result section needs to be re-organized in way it befits for a research paper, explaining the crystal-clear results ONLY. Say for example line 226 to almost 239 (except 236 and 237), is superfluous (“largest delta on Earth with bountiful rivers draining into the northeastern part of the Indian Ocean by way of the Bay of Bengal”) or part of M & M. There are plenty of instances where information fit for ‘M & M’ are written in ‘Results’ like 245-247, 254-257, 259-265, 327-337 and so on).

Response: Previous lines 226 to 239 have been mostly omitted. Previous lines 245-247 were changed. Previous lines 254-257 were kept as is since these are direct rationales that stem from the population study leading into the GWAS study. Previous lines 259-265 have been modified. Previous lines 327-337 have also been modified.

Extensive rewrites have been carried out on the results section and we hope that the reviewer will have a better impression of the section in this manuscript.

Comment: Line 274 – 276: Why it is so, please explain: “Narrow-sense heritability could not be calculated from additive marker effects because the grand mean of the phenotypes could explain more variance than the mixed.solve prediction model”

Response: The output of the prediction model explains less variation than the grand mean. We generally compare any model to the mean to find out how good it is. Since the difference in variation in this case is negative, it could not be done (or would not make statistical sense to do). We have added the following in lines 242-244 in section 3.2 assuming why this might be the case: “This could be attributable to the relatively small sample size with diverse and heterozygous wildtype genotypes used for genomic prediction.”

Comment: Line 306: What is bulky seed in term of their shape and size? Heard for the first time.

Response: All seed related data and results were omitted as per the suggestions of reviewer #1.

Comment: Line 361: Authors need to explain simply that what novel QTLs/ association they found and what was their effect. Their relation with candidate genes can be explained in next section. Could not find the information about the novel QTL in this section (QTL mapping) which could be the most important basis for acceptance of article for publication.

Response: Discovered QTLs have been summarized in table 2. The authors feel that in a properly formatted, published version where the table is viewed right below the text, it would be easy and much more comprehensible. 

Comment: Line 377: What is trait like seed height? I have heard about seed length but not seed height? Are they same? If they are same, then why mentioned as “We previously found seed length to have no correlation with seed height” (line 607-698).

Response: We acknowledge that seed height is an unusual term. It had been explained previously in lines 125 and 126 of the first version. But it is no longer relevant since these all have been removed.

Comment: Line 468-475: This can go as the legend or the footnote of the table.

Response: The table has been removed on the basis of the feedback from the reviewers.

Comment: Line 481-487: “Individual differences in plants of identical genotype are disturbingly disorderly. We found that variance within genotypes in our growth experiments were distinctly large for most traits other than seed properties. But such is customary to plant breeding experiments and many known and unknown factors are in play here. The question here is not how we can remove these confounding variables, since it is not only impossible but also contra-intuitive for practical purposes, but rather how we can derive informative features from observable phenotypes.”. What authors want to convey is not clear. The message should be simple and clear. ‘Individual differences in plants of identical genotype’ : It is not new when you are working with landraces. I am fail to understand the message out of this.

Response: Section was removed.

Comment: Line 493 – 494: This is well establish fact, what is new in this?

Response: Section was removed.

Comment: Line 574: Heading indicate the table number (“More candidate genes in table 4”)? Can’t be some better heading?

Response: The authors are embarrassed about the unintelligent selection of the section title. It has been modified in line 451 to “Discovered QTLs evince new potential gene roles”

Comment: Line 627-628: Concluding a gene/QTL/marker effect, should require concrete experimental results. Just annotation based on literature does not make proper sense. Yes, you can relate but concluding is different and should be based on sound demonstrated results. Same for the functional role for the protein (where are results ? – line 630-631).

Response: We completely agree with the concerns of the reviewer. We have only referred to our results as ‘potential’ genes of interest based on the original association signal in the QTL and some downstream analysis. We have only included these sections as some exposition about the genes that our QTLs of interest house are of relevance. We had acknowledged this even in the first version of the manuscript in the conclusions in lines 658-659: 

“We affirm that we must be careful about the immediate interpretations of our findings as they have not yet been subjected to systematic case-control studies after gene cloning.”

We have made changes throughout the manuscript explicitly mentioning that our genes might ‘potentially’ have such roles. E.g. Line 33, 70, 451, 461, 465, 473, 480 and 486. It was never our intention to claim more than our findings, the functional roles were brought up simply to put forward the descriptive information from the annotations, expression studies and SNP consequences.

The exploration of population structure, correlation between different phenotypes of different groups, the discovered QTLs and the genes of interest from all the mapped genes are the primary findings of this study. On top of this, the git repository is an important asset for researches doing population structure and GWAS studies as all our methods have been elaborately documented there.

 

References:

1. Formentin, E., et al., Transcriptome and Cell Physiological Analyses in Different Rice Cultivars Provide New Insights Into Adaptive and Salinity Stress Responses. Frontiers in Plant Science, 2018. 9(204).

2. Li, F., et al., Genetic Basis Underlying Correlations Among Growth Duration and Yield Traits Revealed by GWAS in Rice (Oryza sativa L.). Frontiers in Plant Science, 2018. 9(650).

3. Yuan, J., et al., Genetic basis and identification of candidate genes for salt tolerance in rice by GWAS. Scientific Reports, 2020. 10(1): p. 9958.

4. Schläppi, M.R., et al., Assessment of Five Chilling Tolerance Traits and GWAS Mapping in Rice Using the USDA Mini-Core Collection. Frontiers in Plant Science, 2017. 8(957).

5. Thapa, R., et al., Genome-Wide Association Mapping to Identify Genetic Loci for Cold Tolerance and Cold Recovery During Germination in Rice. Frontiers in Genetics, 2020. 11(22).

6. Ma, X., et al., Genome-Wide Association Study for Plant Height and Grain Yield in Rice under Contrasting Moisture Regimes. Frontiers in Plant Science, 2016. 7(1801).

7. Wang, X., et al., New Candidate Genes Affecting Rice Grain Appearance and Milling Quality Detected by Genome-Wide and Gene-Based Association Analyses. Frontiers in Plant Science, 2017. 7(1998).

8. Volante, A., et al., Genome-Wide Analysis of japonica Rice Performance under Limited Water and Permanent Flooding Conditions. Frontiers in Plant Science, 2017. 8(1862).

9. Zhao, M., et al., Mining Beneficial Genes for Aluminum Tolerance Within a Core Collection of Rice Landraces Through Genome-Wide Association Mapping With High Density SNPs From Specific-Locus Amplified Fragment Sequencing. Frontiers in Plant Science, 2018. 9(1838).

10. Pariasca-Tanaka, J., C. Baertschi, and M. Wissuwa, Identification of Loci Through Genome-Wide Association Studies to Improve Tolerance to Sulfur Deficiency in Rice. Frontiers in Plant Science, 2020. 10(1668).

11. Crowell, S., et al., Genome-wide association and high-resolution phenotyping link Oryza sativa panicle traits to numerous trait-specific QTL clusters. Nature Communications, 2016. 7(1): p. 10527.

12. Amin, M., et al., Over-expression of a DEAD-box helicase, PDH45, confers both seedling and reproductive stage salinity tolerance to rice (Oryza sativa L.). Molecular Breeding, 2012. 30(1): p. 345-354.

13. Gregorio, G., D. Senadhira, and R. Mendoza, Screening rice for salinity tolerance, vol 22, IRRI discussion paper series. International Rice Research Institute, 1997.

14. Huang, M., et al., BLINK: a package for the next level of genome-wide association studies with both individuals and markers in the millions. GigaScience, 2019. 8(2).

---

## [Decision Letter · Decision Letter 1]

18 Aug 2021

PONE-D-21-14535R1

Novel QTLs for salinity tolerance revealed by genome-wide association studies of biomass, chlorophyll and tissue ion content in 176 rice landraces from Bangladesh

PLOS ONE

Dear Dr. Seraj,

Thank you for submitting your manuscript to PLOS ONE. After careful consideration, we feel that it has merit but does not fully meet PLOS ONE’s publication criteria as it currently stands. Therefore, we invite you to submit a revised version of the manuscript that addresses the points raised during the review process.

Although manuscript has been improved significantly, there are still issues pointed by the Reviewer 3 which need to be addressed. Specifically, candidate genes selection based on RNA-Seq data need careful attention.

We look forward to receiving your revised manuscript.

Kind regards,

Prasanta K. Subudhi, Ph.D.

Academic Editor

PLOS ONE

Additional Editor Comments (if provided):

Major revision

Reviewers' comments:

Reviewer's Responses to Questions

**Comments to the Author**

1. If the authors have adequately addressed your comments raised in a previous round of review and you feel that this manuscript is now acceptable for publication, you may indicate that here to bypass the “Comments to the Author” section, enter your conflict of interest statement in the “Confidential to Editor” section, and submit your "Accept" recommendation.

Reviewer #1: (No Response)

Reviewer #3: (No Response)

2. Is the manuscript technically sound, and do the data support the conclusions?

Reviewer #1: Yes

Reviewer #3: Partly

3. Has the statistical analysis been performed appropriately and rigorously? 

Reviewer #1: Yes

Reviewer #3: No

4. Have the authors made all data underlying the findings in their manuscript fully available?

Reviewer #1: Yes

Reviewer #3: Yes

5. Is the manuscript presented in an intelligible fashion and written in standard English?

Reviewer #1: Yes

Reviewer #3: Yes

6. Review Comments to the Author

Reviewer #1: The manuscript by Alam et al. now titled “Novel QTLs for salinity tolerance revealed genome-wide association studies of biomass, chlorophyll and tissue ion content in 176 rice landraces from Bangladesh” is a revised version of a previously submitted paper. I describes a GWAS mapping analysis of 11 phenotypes yielding 13 QTL and list of associated candidate genes.

The manuscript has been substantially edited and most of my concerns were addressed. However, there are still issues that need to be addressed to improve the quality of this manuscript.

Line 28: take out “sophisticated”. This is a qualified personal opinion and not a quantified fact. There are still a number of such embellishing qualifiers that need to be removed (see below).

Line 63: take out “elaborately”. This is an opinion.

Line 68: edit to “…publicly available gene expression profiles….”, because you did not do this for the current paper.

Line 211: take out “elaborate” (see comments above).

Lines 235-238: use past tense, not present tense, to describe your results.

Lines 252-253: change sentence to “We observed better trait correlations under salt stress conditions than control conditions”. The “fortification” term is too confusing.

Lines 264-28): I still think this paragraph is too long and has unnecessary information. Simply state that non-Gaussian distributions were normalized before GWAS analyses.

Line 281: remove “extensive”. Again, this is an unnecessary qualification/personal opinion.

Line 292: spell out CDP (Comprehensive Data for Plants). Actually, my recommendation is to change the naming to something reflecting salts stress since CDP is a very idiosyncratic.

Lines 295-296: remove the sentence starting with “The suggestive association…”. There is no follow-up to the statement.

Line 304: change “connotations” to “evaluations”.

Lines 330-331: change “imparts sufficient prospect in the role…” to “suggest a role..”.

Line 241: change “To select a set of informative candidate genes with authentic functional roles..” to simply “To select a set of candidate genes…”. The rest are unnecessary qualifications.

Line 425: change “traditionalist” to “traditional”.

Line 428: remove “deliberately”.

Lines 431-433: modify the sentence starting with “The lack of…” to “The lack of this correlation in shoots under stress conditions suggests that sodium exclusion from rice leaves might be critical for survival in salt stress”. At the point, a correlation is not an “elegant demonstration” without further analyses to “demonstrate” causation.

Line 437: remove “highly”.

Line 457: change “causing a nonsense consequence” to “generating a stop codon”.

Line 477: remove the first part of the sentence (“With rigorous….significant thresholds”) and start sentence simply with “We mapped…”.

Line 494: remove “greatly”.

Reviewer #3: The paper has merit and acceptable for publication. However, it needs major revision and should be shortened to focus on QTL detection for salinity tolerance by GWAS. The candidate gene extraction by outsourced RNASeq data is highly speculative in relation to QTLs detected in this study.

RNAseq Data were outsourced from MSU, while Vialone Nano and Baldo RNAseq data were from Formentin 2018. Gene expression and downstream analysis is not appropriate to be included in this paper because RNA gene expression is 1.) genotype-specific, 2.) tissue-specific, 3.) developmental stage-specific, and most importantly, 4.) treatment-specific. While you may have the same ID or variety, the available RNA expression data is only specific to the materials they tested during that time. Moreover, RNA expression studies needs further validation because RNAs are unstable and transient naturally. For this study, to outsource RNASeq data is not acceptable because even MSU RNA Seq data were not collected during salinity stress of the test samples, but mostly during developmental stage of rice and tissue-specific.

Scientific papers are inherently difficult to read and authors should strive to make it more comprehensible by using more simple words to encourage readers. For example line 253-261 states “Rectangle C1 and S1 compare correlations between biomass traits in control and stress conditions, C2 and S2 compare correlations between chlorophyll, SES and biomass in control and stress conditions. C3 and S3 compare correlations between chlorophyll and SES in control and stress. The improved linear correlations between biomass, chlorophyll and SES traits imply that the encumbrance of salt stress on a plant prevents the disproportionate gain of biomass. Since plant biomass and chlorophyll content is markedly reduced under salt stress (supplementary table 3), we can conclude that salt stress does not affect all phenotypes for all genotypes at the same rate and the genetic advantages that any genotype has in terms of tissue growth (root/shoot) or chlorophyll accumulation will be compromised by the effect of abiotic stress at a greater magnitude than traits which have near-baseline values.”

The correlations of traits do not mean causation or effects, it is simply telling a trend between the two traits. Why is such a long confusing conclusion here? What exactly the authors would like to convey based on correlations? It’s a fact that traits like biomass and chlorophyll content are affected by salt stress! The paper would be more meaningful if they measured the reduction of traits in control and stressed genotypes. It would be interesting to find out which varieties had been least affected by the stress and by how much, compare to other genotypes, or an analyses like some sort of index of growth reduction comparison among genotypes with regards to stress and control treatments.

The QTL naming is not reflecting of the association of a trait to a loci. The CDP is too broad and not meaningful to associate to salinity tolerance or any specific trait.

In Discussions, “Biomass traits and chlorophyll content could be valuable indices for the screening of salinity tolerance”—this is already known fact.

Please state the criticism/problem to SES that needs to be addressed. Line 407-408 is intriguing or dramatic and maybe unnecessary to be written.

In discussion, again, high and low correlations are not causation or effects but simply statistical trends. It should also be noted if the correlations are negative or positive. Authors should refrain from drawing too much generalization or speculations unless supported with hard proof validation especially that this study is dealing with diverse germplasms.

Discussion 4.1 is too general and mostly known fact by previous studies. The paper would be more meaningful if authors discussed the traits of varieties used in response to salt stress. This study is nothing new except for the varieties used, so it would be nice to have those information available to the readers in comparison to other known salinity tolerance study.

The paper is claiming novel QTLs, therefore, the focus of discussion should be those novel QTLs, their significance, effects- positive or negative effect, enhancing tolerance or sensitivity, variance explained by the QTLs, occurrence, and usefulness in future salt tolerance introgression. Authors may further discuss the similarity and differences of detected QTLs in relation to previous other QTLs.

In Discussion 4.2, Lines 452-455, candidate genes discussed in the paper were selected on the basis of (a) functionally impairment of gene in the population and (b) individuals carrying the genotype of the functional allele are quantifiably different in terms of phenotype. The study would be meaningful and therefore has merit to discuss candidate genes in details if candidate genes were re-sequenced to confirm the functional SNPs, and validated by gene expression like qRT-PCR. At this point, candidate genes as underlying genes controlling salinity tolerance is premature assumptions despite employing extensive statistical analysis. Gene expression is study-specific and should be tested in contrasting genotypes.

The paper is too lengthy for the methods, and discussion was too short to emphasize the actual findings of the study. While extensive data mining was conducted, the claims regarding candidate genes are not fully supported by current findings. Further validation is needed. However, if the paper is shorted into QTL detection alone, the study is meritorious of publication. I suggest to remove the candidate gene mining by RNASEq, unless fully supported by re-sequencing, functional SNP study and gene expression analysis. We uphold to scientific and rigorous standard regardless of negative or positive results. It is no excuse to say that this study started out as a student project and that it is time constraints and with limited budget. However, once published, this paper would stand as scholastic achievement and pride of the student and all authors included.

7. PLOS authors have the option to publish the peer review history of their article (what does this mean?). If published, this will include your full peer review and any attached files.

Reviewer #1: No

Reviewer #3: No

---

## [Author Response · Author response to Decision Letter 1]

24 Sep 2021

Reviewer #1: 

Comment: The manuscript by Alam et al. now titled “Novel QTLs for salinity tolerance revealed genome-wide association studies of biomass, chlorophyll and tissue ion content in 176 rice landraces from Bangladesh” is a revised version of a previously submitted paper. I describes a GWAS mapping analysis of 11 phenotypes yielding 13 QTL and list of associated candidate genes.

The manuscript has been substantially edited and most of my concerns were addressed. However, there are still issues that need to be addressed to improve the quality of this manuscript.

Line 28: take out “sophisticated”. This is a qualified personal opinion and not a quantified fact. There are still a number of such embellishing qualifiers that need to be removed (see below).

Response: Removed in line 28.

Comment: Line 63: take out “elaborately”. This is an opinion.

Response: Removed in line 62.

Comment: Line 68: edit to “…publicly available gene expression profiles….”, because you did not do this for the current paper.

Response: Added in line 67.

Comment: Line 211: take out “elaborate” (see comments above).

Response: Section has been removed.

Comment: Lines 235-238: use past tense, not present tense, to describe your results.

Response: New lines 233-234 have been changed to past tense.

Comment: Lines 252-253: change sentence to “We observed better trait correlations under salt stress conditions than control conditions”. The “fortification” term is too confusing.

Response: Changed in line 250.

Comment: Lines 264-28): I still think this paragraph is too long and has unnecessary information. Simply state that non-Gaussian distributions were normalized before GWAS analyses.

Response: We have kept the information about data distributions, the test for normalcy and rationale for transformation. We have removed everything else. The new section is in lines 260-273.

Comment: Line 281: remove “extensive”. Again, this is an unnecessary qualification/personal opinion.

Response: Removed in line 292.

Comment: Line 292: spell out CDP (Comprehensive Data for Plants). Actually, my recommendation is to change the naming to something reflecting salts stress since CDP is a very idiosyncratic.

Response: The original dataset generated in our laboratory, from which data was used to prepare the first version of this manuscript, has been titled as such. The authors plan to carry out more GWAS studies with their data using the computation pipeline that has been established in this manuscript. The authors wish to maintain this naming convention for the sake of consistency and convenience.

Comment: Lines 295-296: remove the sentence starting with “The suggestive association…”. There is no follow-up to the statement.

Response: Sentence was removed in line 307.

Comment: Line 304: change “connotations” to “evaluations”.

Response: The authors feel that the word ‘connotations’ is more appropriate here.

Comment: Lines 330-331: change “imparts sufficient prospect in the role…” to “suggest a role..”.

Response: Changed in line 344.

Comment: Line 241: change “To select a set of informative candidate genes with authentic functional roles..” to simply “To select a set of candidate genes…”. The rest are unnecessary qualifications.

Response: Section has been moved to supplementary notes. So, left unchanged.

Comment: Line 425: change “traditionalist” to “traditional”.

Response: Changed in line 377.

Comment: Line 428: remove “deliberately”.

Response: Removed in line 380.

Comment: Lines 431-433: modify the sentence starting with “The lack of…” to “The lack of this correlation in shoots under stress conditions suggests that sodium exclusion from rice leaves might be critical for survival in salt stress”. At the point, a correlation is not an “elegant demonstration” without further analyses to “demonstrate” causation.

Response: The authors concede. The change has been made in lines 384-285.

Comment: Line 437: remove “highly”.

Response: Changed in line 389.

Comment: Line 457: change “causing a nonsense consequence” to “generating a stop codon”.

Response: Section has been moved to supplementary. 

Comment: Line 477: remove the first part of the sentence (“With rigorous….significant thresholds”) and start sentence simply with “We mapped…”.

Response: Changed in line 426.

Comment: Line 494: remove “greatly”.

Response: Changed in line 443.

Reviewer #3: 

Comment: The paper has merit and acceptable for publication. However, it needs major revision and should be shortened to focus on QTL detection for salinity tolerance by GWAS. The candidate gene extraction by outsourced RNASeq data is highly speculative in relation to QTLs detected in this study.

Response: We appreciate the new perspectives and valuable insights from the reviewer. In this revision, we omit outsourced expression studies from the main manuscript and add more detail about inter-variety variations in Bangladeshi landraces in response to salt stress. 

Comment: RNAseq Data were outsourced from MSU, while Vialone Nano and Baldo RNAseq data were from Formentin 2018. Gene expression and downstream analysis is not appropriate to be included in this paper because RNA gene expression is 1.) genotype-specific, 2.) tissue-specific, 3.) developmental stage-specific, and most importantly, 4.) treatment-specific. While you may have the same ID or variety, the available RNA expression data is only specific to the materials they tested during that time. Moreover, RNA expression studies needs further validation because RNAs are unstable and transient naturally. For this study, to outsource RNASeq data is not acceptable because even MSU RNA Seq data were not collected during salinity stress of the test samples, but mostly during developmental stage of rice and tissue-specific.

Response: The reviewer raises some valid concerns. We agree that the impact of the manuscript up to QTL mapping is far greater than its remainder which yields speculative conclusions at best. We have streamlined the manuscript further to accommodate the views of the reviewer. Instead of completely deleting the expression analysis and functional consequence studies, we have moved the complete section to supplementary notes should any keen reader be interested in going over them.

Comment: Scientific papers are inherently difficult to read and authors should strive to make it more comprehensible by using more simple words to encourage readers. For example line 253-261 states “Rectangle C1 and S1 compare correlations between biomass traits in control and stress conditions, C2 and S2 compare correlations between chlorophyll, SES and biomass in control and stress conditions. C3 and S3 compare correlations between chlorophyll and SES in control and stress. The improved linear correlations between biomass, chlorophyll and SES traits imply that the encumbrance of salt stress on a plant prevents the disproportionate gain of biomass. Since plant biomass and chlorophyll content is markedly reduced under salt stress (supplementary table 3), we can conclude that salt stress does not affect all phenotypes for all genotypes at the same rate and the genetic advantages that any genotype has in terms of tissue growth (root/shoot) or chlorophyll accumulation will be compromised by the effect of abiotic stress at a greater magnitude than traits which have near-baseline values.”

The correlations of traits do not mean causation or effects, it is simply telling a trend between the two traits. Why is such a long confusing conclusion here? What exactly the authors would like to convey based on correlations? It’s a fact that traits like biomass and chlorophyll content are affected by salt stress! The paper would be more meaningful if they measured the reduction of traits in control and stressed genotypes. It would be interesting to find out which varieties had been least affected by the stress and by how much, compare to other genotypes, or an analyses like some sort of index of growth reduction comparison among genotypes with regards to stress and control treatments.

Response: The reviewer is correct that correlations of traits do not explicitly mean causation or effects but we disagree with their opinion that correlation is ‘simply a trend’. The word ‘trend’ in statistics is more appropriate for fields like economics where only directional patterns are observed. 

We want to take this opportunity to break down the section quoted here by the reviewer. To make things simple, let us look at the correlation between Shoot weight and Chlorophyll A in control and stress conditions. Please download the notebook at: https://github.com/DeadlineWasYesterday/Cat-Does-Plant/blob/master/Figures%20and%20tables/Revision%202%20argument.pdf to follow this demonstration. There should be a download button on the top right of the preview in this webpage. 

Please look at figure 1 in this notebook. It shows that shoot weight is reduced under salt stress. Figure 2 shows a less dramatic, but identifiable reduction in chlorophyll A content. From this, we can conclude that shoot weight and chlorophyll A content is reduced under salt stress. Figures 3 and 4 linearly model the nature of this reduction. We can observe a clear linear relationship under stress conditions which was less prominent in the absence of salt. This is what the color depth of the rectangles in figure 3 of our manuscript signifies. 

In figure 5 we have plotted the distribution of the ratio of shoot weight and chlorophyll A content from supplementary table 3. This shows that both the magnitude and the spread of the shoot weight to chlorophyll A ratio is reduced in stress conditions. The bidirectional bar chart found in the last figure (out[10]), shows a dramatic difference between the dispersal of shoot weight to chlorophyll A ratios in our experimental varieties. From this bar chart, we can see that plants that had a high ratio in control conditions, exhibit smaller ratios in stress conditions. 

This is the exact conclusion we have drawn in the section quoted from the manuscript by the reviewer. The correlation heatmap on figure 3, the p values in supplementary table 5 and the density plots in supplementary figure 2 directly demonstrate this phenomenon. To paraphrase: Salt does not affect all genotypes equally. In salt stress, phenotypes correlate better linearly and the ratios become more proportional, which is to say that any advantage that a genotype has in terms of disproportionate tissue growth or chlorophyll accumulation will be compromised first under salt stress in general.

We beg the reviewer to not undermine our analyses by calling them ‘merely a trend’. A sufficient amount of thought and attention was put into this. The authors feel that there is ample data in the manuscript to support their claim. 

We have added lines 263-275 and a new figure 4 to bring forward the effects of salt stress on phenotypes of different varieties. 

Comment: The QTL naming is not reflecting of the association of a trait to a loci. The CDP is too broad and not meaningful to associate to salinity tolerance or any specific trait.

Response: The rationale behind the original naming convention was that this was a diverse set of phenotype data for GWAS analysis. We had filtered out quite a few before writing the original manuscript and as per the recommendations of the reviewers, traits other than those concerned with salinity tolerance were left out from the first revision. The naming is consistent throughout the manuscript, images and supplementary files and we hope to keep them consistent in our following work using data that have been excluded from this manuscript. Additionally, the QTL names with qCDP are hardcoded in all the sourcecode, making our experiments easily reproducible using data that was made available online. 

Comment: In Discussions, “Biomass traits and chlorophyll content could be valuable indices for the screening of salinity tolerance”—this is already known fact.

Response: There are a number of recent studies on salinity tolerance where the experimental design exclusively focuses on ion content and ion ratios and/or overlooks simpler observations such as biomass and chlorophyll content [1-3]. The authors agree that the effects of salt stress on biomass and chlorophyll is quite obvious. In spite of this, our study as a whole brings forward the numerical relationships between these phenotypes in response to salinity tolerance. The quoted section in particular reflects the basic impression of our journey throughout the study. A lot of it builds on the referenced manuscript by Pires et al. 2015 [4]. The authors wish to keep the section as is. 

Comment: Please state the criticism/problem to SES that needs to be addressed. Line 407-408 is intriguing or dramatic and maybe unnecessary to be written.

Response: SES is a composite score which is prone to human error. The principal component we derived from stress biomass and chlorophyll content in this revision would be more numerically robust and a better index for salt injury than an evaluation made by the naked eye. Of course, SES scoring is less laborious and still applicable in many scenarios. The authors feel that a comprehensive study of salt tolerance will still be incomplete without recording biomass and chlorophyll content. In light of this, we have appended the following section at the end of our conclusions in line 435: “Although it is well established that biomass and chlorophyll related traits alongside tissue ion content aids the assessment of salinity tolerance, our studies show that the numerical values of plant biomass and chlorophyll content could manifest themselves into a more objective index of salt injury unabated by human error.”

Comment: In discussion, again, high and low correlations are not causation or effects but simply statistical trends. It should also be noted if the correlations are negative or positive. Authors should refrain from drawing too much generalization or speculations unless supported with hard proof validation especially that this study is dealing with diverse germplasms.

Response: The blue color in figure 3 and supplementary figure 1 indicates a positive correlation and the red color indicates negative. Out of the significant correlations shown in figure 3, only SES shares negative correlation with the remaining phenotypes. The authors have addressed part of this comment in detail in the response to the second comment.

Comment: Discussion 4.1 is too general and mostly known fact by previous studies. The paper would be more meaningful if authors discussed the traits of varieties used in response to salt stress. This study is nothing new except for the varieties used, so it would be nice to have those information available to the readers in comparison to other known salinity tolerance study.

Response: The authors feel differently about section 4.1. We feel that the contents of the section have more depth than the reviewer has surmised here. The distributions of our phenotypes have been observed. We have discussed why we did not derive phenotypes from sodium and potassium ratios, which are very commonly used in the screening of salinity tolerance. We feel that our study puts forward ample evidence to allow us to emphasize the value of biomass and chlorophyll data in the study of salinity tolerance. 

Comment: The paper is claiming novel QTLs, therefore, the focus of discussion should be those novel QTLs, their significance, effects- positive or negative effect, enhancing tolerance or sensitivity, variance explained by the QTLs, occurrence, and usefulness in future salt tolerance introgression. Authors may further discuss the similarity and differences of detected QTLs in relation to previous other QTLs.

Response: This is primarily a QTL detection and mapping study. Molecular dissection of these QTLs requires thorough and extensive laboratory work, which the authors have been planning for the near future. The topics brought up by the reviewer are very relevant and will be the subject of our subsequent studies.

Comment: In Discussion 4.2, Lines 452-455, candidate genes discussed in the paper were selected on the basis of (a) functionally impairment of gene in the population and (b) individuals carrying the genotype of the functional allele are quantifiably different in terms of phenotype. The study would be meaningful and therefore has merit to discuss candidate genes in details if candidate genes were re-sequenced to confirm the functional SNPs, and validated by gene expression like qRT-PCR. At this point, candidate genes as underlying genes controlling salinity tolerance is premature assumptions despite employing extensive statistical analysis. Gene expression is study-specific and should be tested in contrasting genotypes.

Response: These sections have been removed in accordance with the comments from the reviewer. 

Comment: The paper is too lengthy for the methods, and discussion was too short to emphasize the actual findings of the study. While extensive data mining was conducted, the claims regarding candidate genes are not fully supported by current findings. Further validation is needed. However, if the paper is shorted into QTL detection alone, the study is meritorious of publication. I suggest to remove the candidate gene mining by RNASEq, unless fully supported by re-sequencing, functional SNP study and gene expression analysis. We uphold to scientific and rigorous standard regardless of negative or positive results. It is no excuse to say that this study started out as a student project and that it is time constraints and with limited budget. However, once published, this paper would stand as scholastic achievement and pride of the student and all authors included.

Response: We have revised the manuscript as per the suggestions of the reviewer.

References:

1. Warraich, A.S., et al., Rice GWAS reveals key genomic regions essential for salinity tolerance at reproductive stage. Acta Physiologiae Plantarum, 2020. 42(8): p. 134.

2. Kumar, V., et al., Genome-wide association mapping of salinity tolerance in rice (Oryza sativa). DNA Research, 2015. 22(2): p. 133-145.

3. Batayeva, D., et al., Genome-wide association study of seedling stage salinity tolerance in temperate japonica rice germplasm. BMC Genetics, 2018. 19(1): p. 2.

4. Pires, I.S., et al., Comprehensive phenotypic analysis of rice (Oryza sativa) response to salinity stress. Physiologia Plantarum, 2015. 155(1): p. 43-54.

---

## [Editor Report · Decision Letter 2]

11 Oct 2021

PONE-D-21-14535R2Novel QTLs for salinity tolerance revealed by genome-wide association studies of biomass, chlorophyll and tissue ion content in 176 rice landraces from BangladeshPLOS ONE

Dear Dr. Seraj,

Thank you for submitting your manuscript to PLOS ONE. After careful consideration, we feel that it has merit but does not fully meet PLOS ONE’s publication criteria as it currently stands. Therefore, we invite you to submit a revised version of the manuscript that addresses the points raised during the review process.

Specifically, following things need to be addressed.1. Since there are some associations between genes and QTLs mentioned in section 3.4 and conclusion section include a sentence 'Supplementary analysis of gene expression and functional annotations assumes

432 potential roles for a number of genes within the identified QTLs', authors are advised to include few lines in the discussion section regarding this. 2. It seems there are some supplementary tables and figures not cited in the text. Please ensure that all supplementary information is cited.

We look forward to receiving your revised manuscript.

Kind regards,

Prasanta K. Subudhi, Ph.D.

Academic Editor

PLOS ONE

Journal Requirements:

Additional Editor Comments (if provided):

Minor revision
---

## [Author Response · Author response to Decision Letter 2]

13 Oct 2021

Comment from the editor: 

1. Since there are some associations between genes and QTLs mentioned in section 3.4 and conclusion section include a sentence 'Supplementary analysis of gene expression and functional annotations assumes potential roles for a number of genes within the identified QTLs', authors are advised to include few lines in the discussion section regarding this. 

2. It seems there are some supplementary tables and figures not cited in the text. Please ensure that all supplementary information is cited.

Response:

1. We have added lines 424 to 452 regarding the expression and functional studies.

2. We have cited all the supplementary information in the text:

Supplementary table 1 cited in line 81.

Supplementary table 2 cited in line 236.

Supplementary table 3 cited in line 237.

Supplementary table 4 cited in line 246.

Supplementary table 5 cited in line 249.

Supplementary table 6 cited in line 283.

Supplementary table 7 cited in line 318.

Supplementary table 8 cited in line 327.

Supplementary table 9 cited in line 329.

Supplementary table 10 cited in line 427.

Supplementary table 11 cited in line 430.

Supplementary table 12 cited in line 432.

Supplementary table 13 cited in line 433.

Supplementary table 14 cited in line 434.

Supplementary figure 1 cited in line 248.

Supplementary figure 2 cited in line 277.

Supplementary figure 3 cited in line 287.

Supplementary figure 4 cited in line 289.

Supplementary figure 5 cited in line 307.

Supplementary figure 6 cited in line 324.

Supplementary figure 7 cited in line 430.

Supplementary text 1 cited in line 426.

---

## [Editor Report · Decision Letter 3]

20 Oct 2021

Novel QTLs for salinity tolerance revealed by genome-wide association studies of biomass, chlorophyll and tissue ion content in 176 rice landraces from Bangladesh

PONE-D-21-14535R3

Dear Dr. Seraj,

We’re pleased to inform you that your manuscript has been judged scientifically suitable for publication and will be formally accepted for publication once it meets all outstanding technical requirements.

Kind regards,

Prasanta K. Subudhi, Ph.D.

Academic Editor

PLOS ONE

Additional Editor Comments (optional):

Accept